# Architectural basis for cylindrical self-assembly governing Plk4-mediated centriole duplication in human cells

Jong Il Ahn [1,12], Liang Zhang[1,12], Harsha Ravishankar [1], Lixin Fan[2], Klara Kirsch [1], Yan Zeng[1], Lingjun Meng[1], Jung-Eun Park [1], Hye-Yeoung Yun[3], Rodolfo Ghirlando [4], Buyong Ma [5,11], David Ball[6], Bonsu Ku [3], Ruth Nussinov[5,7], Jeremy D. Schmit [8], William F. Heinz [9], Seung Jun Kim[3], Tatiana Karpova [6], Yun-Xing Wang [10] & Kyung S. Lee [1✉]

Proper organization of intracellular assemblies is fundamental for efficient promotion of biochemical processes and optimal assembly functionality. Although advances in imaging technologies have shed light on how the centrosome is organized, how its constituent proteins are coherently architected to elicit downstream events remains poorly understood. Using multidisciplinary approaches, we showed that two long coiled-coil proteins, Cep63 and Cep152, form a heterotetrameric building block that undergoes a stepwise formation into higher molecular weight complexes, ultimately generating a cylindrical architecture around a centriole. Mutants defective in Cep63•Cep152 heterotetramer formation displayed crippled pericentriolar Cep152 organization, polo-like kinase 4 (Plk4) relocalization to the procentriole assembly site, and Plk4-mediated centriole duplication. Given that the organization of pericentriolar materials (PCM) is evolutionarily conserved, this work could serve as a model for investigating the structure and function of PCM in other species, while offering a new direction in probing the organizational defects of PCM-related human diseases.

[1] Cancer Innovation Laboratory, Center for Cancer Research, National Cancer Institute, National Institutes of Health, Bethesda, MD 20892, USA. [2] Basic Science Program, Frederick National Laboratory for Cancer Research, Small-Angle X-ray Scattering Core Facility, National Cancer Institute, National Institutes of Health, Frederick, MD 21702, USA. [3] Disease Target Structure Research Center, Korea Research Institute of Bioscience and Biotechnology, Daejeon, Republic of Korea. [4] Laboratory of Molecular Biology, National Institute of Diabetes and Digestive and Kidney Diseases, National Institutes of Health, Bethesda, MD 20892, USA. [5] Basic Science Program, Leidos Biomedical Research, Inc., Cancer and Inflammation Program, National Cancer Institute, Frederick, MD 21702, USA. [6] Laboratory of Receptor Biology and Gene Expression, Optical Microscopy Core, National Cancer Institute, National Institutes of Health, Bethesda, MD 20892, USA. [7] Department of Human Molecular Genetics and Biochemistry, Sackler School of Medicine, Tel Aviv University, Tel Aviv 69978, Israel. [8] Department of Physics, Kansas State University, Manhattan, KS 66506, USA. [9] Optical Microscopy and Analysis Laboratory, Cancer Research Technology Program, Frederick National Laboratory for Cancer Research, Frederick, MD 21702, USA. [10] Protein-Nucleic Acid Interaction Section, Center for Structural Biology, National Cancer Institute, National Institutes of Health, Frederick, MD 21702, USA. [11] Present address: School of Pharmacy, Shanghai Jiao Tong University, 200240 Shanghai, P R China. [12] These authors contributed equally: Jong Il Ahn, Liang Zhang. ✉email: kyunglee@mail.nih.gov

Research shows that a cell is hierarchically organized from single molecules to higher-order assemblies to larger organelles. However, how the constituent components of a higher-order entity work coherently to elicit biological process(es) remains largely uninvestigated. As an electron-dense, membraneless organelle, the centrosome plays a critical role in promoting various cellular events, including cell division, cell motility, and intracellular signaling[1,2]. The centrosome is composed of a pair of orthogonally arranged, microtubule (MT)-derived apparatus, called centrioles, and their surrounding pericentriolar material (PCM) packed with a few hundred copies of several hundred different proteins[3]. Subdiffraction-resolution imaging of centrosomes in flies and humans reveals that the constituent components of interphase PCM are highly organized into concentric layers of cylindrical localization patterns around a centriole[4–7]. The centriole is thought to be central in organizing inner PCM layers, although a centriole-independent interphase PCM assembly has also been reported[8]. Yet, how these PCM layers are structured at the molecular level to place other PCM proteins and promote their functions remains largely unknown[9–11].

Studies have shown that, in humans, two long coiled-coil (CC) proteins, Cep63 and Cep152, which cooperatively generate an antiparallelly arranged 2:2 heterotetrameric complex[12], localize around a centriole and promote centriole duplication during the G1/S phase of the cell cycle[13–16]. Remarkably, Cep63 and Cep152 can self-assemble into a radially aligned, submicrometer-scale, cylindrical structure in vitro in such a fashion that the N-termini of Cep63 and Cep152 are placed at the innermost and outermost regions, respectively, of the structure[12]. The dimensional and spatial organization of the in vitro Cep63•Cep152 self-assembly are comparable to the pericentriolar localization patterns of endogenous Cep63 and Cep152[17–19]. These findings indicate that, unlike other structural CC proteins, such as vimentin and keratin, which form elongated, cable-like filaments for physical strength, Cep63 and Cep152 possess distinct physicochemical properties that enable them to generate higher molecular weight (MW) complex(es) and construct a cylindrical self-assembly around a centriole. Moreover, mutations in Cep63 or Cep152 that disrupt Cep63•Cep152 self-assembly result in impaired centriole duplication, which could lead to chromosome missegregation and yield genetic disorders such as cancer, microcephaly, ciliopathy, and dwarfism[20,21]. Yet, the molecular mechanisms underlying these processes remain largely unknown because of the difficulty of studying multi-step, higher-order assembly processes.

Proper organization of the PCM space, which would underpin the spatial placement of PCM-resident proteins, could exert a profound effect on the molecular behavior and functionality of these proteins. Various studies demonstrated that polo-like kinase 4 (Plk4) is a key regulator of centriole biogenesis[22–24], and that its level autonomously oscillates at the base of the growing centriole[25]. Plk4 binds to the N-terminal motif of Cep152 via its C-terminal cryptic polo-box[13,14], consequently placing itself at the outskirts of a cylindrically localized Cep152 placed around a centriole[12,16]. Plk4 then undergoes *trans*-autophosphorylation[26]-dependent liquid-liquid phase separation (LLPS) and relocalization from around Cep152 (ring-like state) to the procentriole assembly site (dot-like state)[27–30]. Other data demonstrate that Plk4's autoactivation-dependent ring-to-dot relocalization is an essential step for inducing centriole biogenesis[29] and the concentration of Plk4 determines the temporal onset of centriole assembly[31]. Therefore, the organizational nature (i.e., dimension and density of the Plk4-binding Cep152 N-terminus) of the cylindrical Cep63•Cep152 platform, which would influence the level of recruited Plk4 and its inter-molecular distance and dynamics, could govern the timing of Plk4's activation and functionality in the three-dimensional (3D) pericentriolar space.

In this study, we combined multidisciplinary approaches to investigate how the two scaffolds, Cep63 and Cep152, self-assemble into a cylindrically organized pericentriolar architecture and how a defect in this process alters Plk4-mediated centriole duplication. Our data show that the Cep63•Cep152 platform is self-assembled through a concentration-dependent, stepwise process, and that this event is dynamic and reversible. Characterization of two Cep152 mutants impaired in self-assembly formation revealed that misorganizing the higher-order Cep63•Cep152 platform results in a reduced Cep152 concentration around a centriole, consequently crippling Plk4's ring-to-dot relocalization and subsequent induction of procentriole formation. Thus, unlike other kinases whose activity is regulated by upstream kinases or regulators, how the Cep63•Cep152 platform is organized predetermines the functionality of Plk4 recruited to the Cep152 tether. Given that the organization of PCM is highly conserved from worms to humans, this work could provide new insights into the fundamentals regulating how the pericentriolar architectures self-organize and function as a holistic entity in various organisms. It could also offer a new direction for investigating the structural and functional abnormalities associated with centrosome-associated human disorders.

## Results

**Spatial organization of the Cep63•Cep152 platform around a centriole.** To gain an overall view of how Cep63 and Cep152 are organized around a centriole, mCherry-Cep63-mGFP (i.e., the full-length Cep63 containing an N-terminally tagged mCherry and a C-terminally tagged mGFP[32]; both mCherry and mGFP are monomeric) and mGFP-Cep152-mCherry constructs were expressed in U2OS cells after depleting the respective endogenous proteins by RNAi. 3D-structured-illumination microscopy (3D-SIM) showed that, in line with earlier findings[12,19], Cep63 and Cep152 displayed distinguishable localization patterns around the longitudinal axis of a centriole within a confined volume of the inner PCM space (Fig. 1a and Supplementary Fig. 1).

Notably, the N-terminus of Cep63 (i.e., the mCherry fluorescence of Cep63) was detected in the innermost region (231 ± 23 nm in diameter), whereas the N-terminus of Cep152 (i.e., the mGFP fluorescence of Cep152) was positioned at the outermost region (350 ± 23 nm in diameter). These observations agree with the previously proposed model that Cep63 and Cep152 generate an antiparallel 2:2 heterotetrameric complex arranged around a centriole in the Cep152 N-terminus-outward fashion[12]. In addition, estimated diameters and heights of cylindrical mCherry and mGFP fluorescence suggested that the total length spans (i.e., from the N-terminus to the C-terminus) for Cep63 and Cep152 reach up to 47 ± 21 nm and 82 ± 26 nm, respectively, and up to ~300 nm in height (Fig. 1b). Hence, given that the Cep63•Cep152 complex self-assembles into a higher-order cylindrical architecture in vitro[12], Cep63 and Cep152, which assume an elongated structure, appear to be able to self-organize into an extended cylindrical platform around a centriole (Fig. 1c). Given Cep152's ability to bind to Cep63 and self-assemble into a higher-order cylindrical structure, these observations suggest that the heterotetrameric complex generated by Cep63 and Cep152 may have innate physicochemical properties that allow it to self-organize around a centriole and achieve optimal functionality.

**The heterotetrameric Cep63•Cep152 complex assumes a multiply bent morphology.** To understand how the tetrameric Cep63•Cep152 complex is assembled around a centriole at the molecular level, we set out to investigate the structural properties of the complex. Cep63 and Cep152 primarily exhibit a predicted CC conformation (Fig. 1c, left), suggesting that they may assume

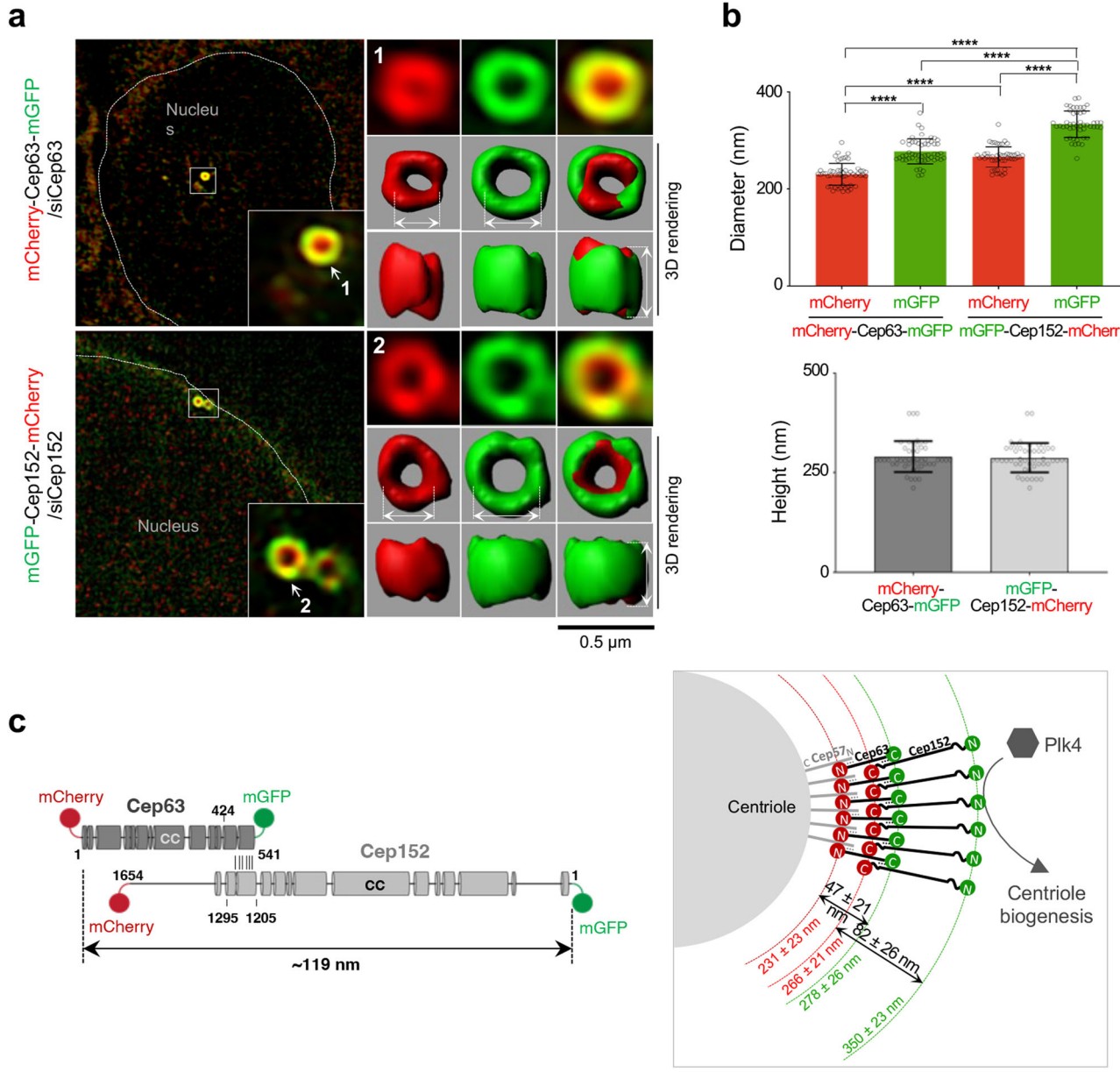

**Fig. 1 The organization of Cep63 and Cep152 at the interphase PCM. a, b** 3D-SIM analyses for U2OS cells stably expressing the indicated Cep63 or Cep152 constructs after depleting respective endogenous Cep63 or Cep152 by RNAi. **a** Representative images displaying the PCM-localized mCherry-Cep63-mGFP or mGFP-Cep152-mCherry signals are shown with two surface-rendered models (top and side views). Original fluorescence images are provided in Supplementary Fig. 1a, b. Boxes, areas of enlargement; double arrows, the diameters and heights measured for quantification. **b** Quantification of the mCherry and mGFP fluorescent signals in (**a**) to determine the peak-to-peak diameters (top) and heights (bottom) for the cylindrically localized mCherry-Cep63-mGFP (total $n = 53$ and $n = 48$, respectively) and mGFP-Cep152-mCherry (total $n = 49$ and $n = 48$, respectively) obtained from three independent experiments. Error bars, mean of $n \pm$ s.d. *$P < 0.05$, ****$P < 0.0001$ (unpaired two-tailed $t$ test). Detailed methods employed for quantification are described in Supplementary Fig. 1c, d. **c** (left) A schematic showing the structures of Cep63 and Cep152 with various lengths of CCs (round bars) predicted by the COILS server[76]. **c** (right) The organization of Cep63 and Cep152 around a centriole, illustrated based on prior observations[12,13,16,27,40,43]. Numbers (red and green) indicate the diameters from (**b**).

a linearly structured, long, rod-like morphology that extends approximately 119 nm from the N-terminus of Cep63 to the N-terminus of Cep152 (Fig. 1c). The Cep63 (424–541)•Cep152 (1205–1295) complex, which contains the entire interaction region between Cep63 and Cep152, bears the full capacity of self-assembling into a higher-order cylindrical architecture[12]. Therefore, to reveal the structural features of this complex, we performed small-angle X-ray scattering (SAXS) analysis at low, non-assembling concentrations (13 and 25 μM) (Supplementary Fig. 2a, b) and reconstructed a shape model ab initio using the

DAMMIN (ATSAS) program[33], in which the model was built from densely packed dummy atoms (as an ensemble of packed beads). A simulated annealing method was applied to maximize the fit between the theoretical scattering calculated from the packed beads and the experimental curve.

The SAXS model for the Cep63 (424–541)•Cep152 (1205–1295) complex exhibited an elongated, yet multiply bent, morphology (Fig. 2a). Although shape reconstructions of elongated bodies (>1:5 anisometry) tend to yield a somewhat bent shape[34], this finding suggests that the complex could be

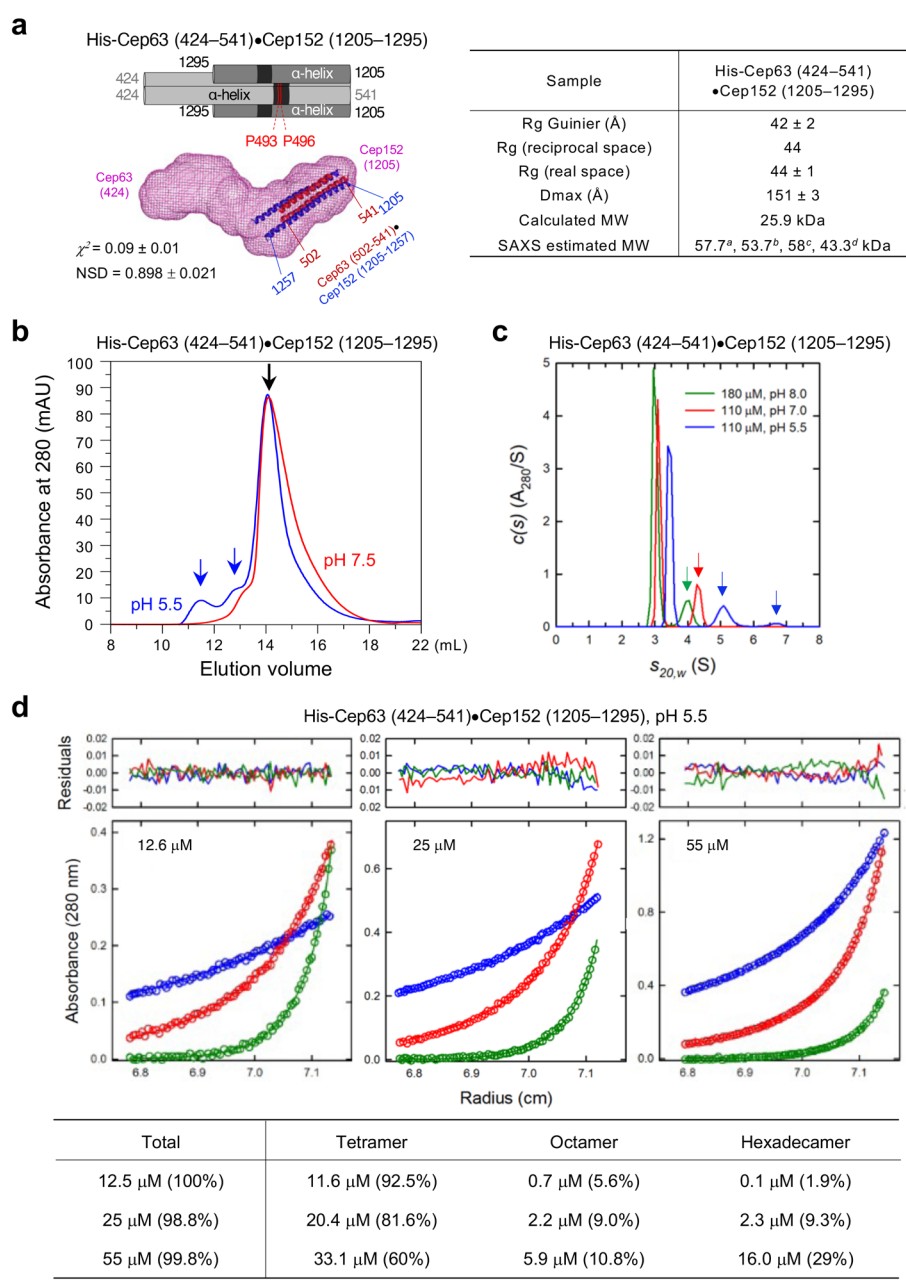

| Sample | His-Cep63 (424–541)•Cep152 (1205–1295) |
|---|---|
| Rg Guinier (Å) | 42 ± 2 |
| Rg (reciprocal space) | 44 |
| Rg (real space) | 44 ± 1 |
| Dmax (Å) | 151 ± 3 |
| Calculated MW | 25.9 kDa |
| SAXS estimated MW | 57.7[a], 53.7[b], 58[c], 43.3[d] kDa |

| Total | Tetramer | Octamer | Hexadecamer |
|---|---|---|---|
| 12.5 µM (100%) | 11.6 µM (92.5%) | 0.7 µM (5.6%) | 0.1 µM (1.9%) |
| 25 µM (98.8%) | 20.4 µM (81.6%) | 2.2 µM (9.0%) | 2.3 µM (9.3%) |
| 55 µM (99.8%) | 33.1 µM (60%) | 5.9 µM (10.8%) | 16.0 µM (29%) |

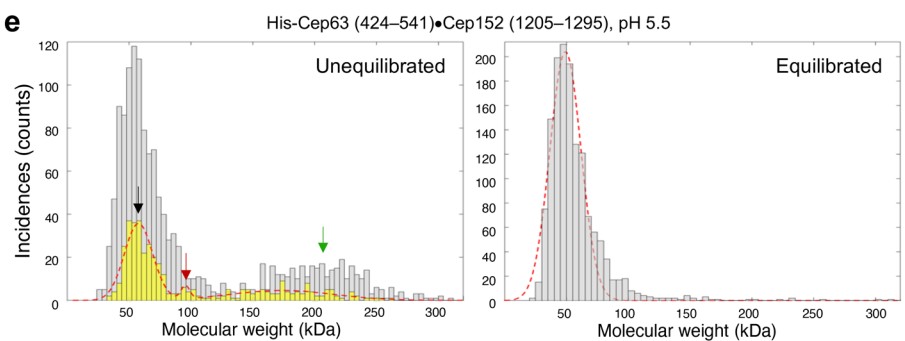

structurally flexible[35]. To help define the orientation of the envelope, additional SAXS analyses were performed with two longer complexes [i.e., Cep63 (220–541)•Cep152 (1140–1295) and Cep63 (219–541)•Cep152 (1205–1295)], which also yielded a characteristic morphological kink (Supplementary Fig. 2c–g). A best-fitting overlay of three envelopes with the docked crystal

structure (red and blue helices) of the Cep63 (502–541)•Cep152 (1205–1250) complex (PDB: 6CSU) (Supplementary Fig. 2h and Supplementary Movie 2) suggested that a structural kinking is present immediately upstream of the Cep63 (502–541) dimer (two red helices). This morphological feature could be attributable to the presence of the two Pro residues (P493 and P496) in

**Fig. 2 A SAXS-derived ab initio envelope for the heterotetrameric Cep63 (424–541)•Cep152 (1205–1295) complex and the cooperative formation of its hexadecameric form. a** A schematic and the SAXS envelope for the heterotetrameric Cep63 (424–541)•Cep152 (1205–1295) complex are shown with the embedded crystal structure of Cep63 (502–541)•Cep152 (1205–1250) (PDB: 6CSU; red and blue helices). Averaged $\chi^2$ (the difference between actual and expected data) and normalized spatial discrepancy (NSD) values calculated from 32 independent DAMMIN reconstructions and $Rc$ (cross-sectional radius) calculated from low $q$ Guinier fit are shown. Various physical parameters of the complex calculated from its respective SAXS curve are provided (table, right). [a,b,c,d], MWs determined from the SAXS data using four different methods (see Methods for details). Raw data are provided in Supplementary Fig. 2a, b. A 3D-rendered envelope is provided as Supplementary Movie 1. **b** SEC of Cep63 (424–541)•Cep152 (1205–1295) performed at the indicated pH. Black arrow, the heterotetrameric complex; blue arrows, higher-MW complexes. **c** Sedimentation velocity $c(s)$ profiles for the Cep63 (424–541)•Cep152 (1205–1295) complex under the indicated conditions. Colored arrows, faster-sedimenting, higher-MW species detected under the respective concentrations. Samples were analyzed in 3 mm pathlength cells. **d** Sedimentation equilibrium absorbance data collected for the same complex in (**c**) at 7,000 (blue), 11,000 (red), and 20,000 (green) rpm at the loading concentrations indicated. Data collected at pH 5.5 were analyzed globally in terms of a tetramer–octamer–hexadecamer reversible self-association model. For clarity, only every third experimental data point is shown. Best fits are represented by a solid line through the experimental points, and the combined residuals are shown above the plots. The table (bottom) shows concentrations for the tetramer, octamer, and hexadecamer (in tetramer units) calculated based on the best-fit reversible self-association model. **e** Histograms showing the particle distribution of the same complex in (**c**) as a function of molecular weight. The y-axis (Incidence) denotes the number of particles. The yellow histogram with the red dotted line was generated by reanalyzing the data after discarding a third of the particles, as described previously[77].

Cep63 (as indicated in Fig. 2a), which could impose a unique conformational constraint on the structure of a peptide chain[36]. An estimated MW (mean of 43.3 kDa) (see Methods) from the SAXS profile (Fig. 2a, right) suggests that Cep63 (424–541)•Cep152 (1205–1295) (calculated MW of 25.9 kDa) forms a dimer of a dimer.

**Concentration-dependent formation of a hexadecameric Cep63•Cep152 complex.** Size-exclusion chromatography (SEC) performed with the Cep63 (424–541)•Cep152 (1205–1295) complex at pH 7.5 showed a single, ~50 kDa peak (Fig. 2b and Supplementary Fig. 2i), consistent with the 2:2 heterotetrameric (i.e., building block) complex state[12]. When SEC was carried out at pH 5.5, the complex generated discrete, higher MW species (Fig. 2b and Supplementary Fig. 2i), suggesting that higher MW complexes can be induced under these conditions.

Sedimentation velocity analysis at pH 8.0 revealed that the Cep63 (424–541)•Cep152 (1205–1295) complex sedimented largely as a 2.83S species (and a small fraction of a fast-sedimenting species, green arrow in Fig. 2c) with an estimated MW of 51 kDa (Fig. 2c), supporting the idea that the complex is a heterotetramer with a 2:2 stoichiometry (calculated MW of 51.8 kDa). In line with the data in Fig. 2b, as the pH decreased to 7 and to 5.5, the amount of the fast-sedimenting species increased (red and blue arrows in Fig. 2c), resulting in the appearance of species up to ~6.6 S (blue arrow). Thus, altering the ionic strength of the solution by pH can significantly influence the degree of inter-heterotetrameric interactions required to generate higher MW complexes.

To further characterize the fast-migrating species at pH 5.5, a series of sedimentation equilibrium analyses were performed with different concentrations of the Cep63 (424–541)•Cep152 (1205–1295) complex, collecting sedimentation data at 7000, 12,000, and 20,000 revolutions per minute (rpm) (Fig. 2d and Supplementary Fig. 2j). The weight-average molar mass obtained increased with the loading concentration, suggesting the presence of a reversible self-association. Rigorous data analyses supported a reversible tetramer–octamer–hexadecamer self-association model (see details in Methods), exhibiting a concentration-dependent increase in the percentage of hexadecamer [i.e., the tetramer of the Cep63 (424–541)•Cep152 (1205–1295) heterotetramer] (Fig. 2d, bottom). Remarkably, the presence of dodecamer was not apparent under all experimental conditions, suggesting cooperativity in generating the hexadecamer. Analyses of the contribution of tetramer (left), octamer (middle), and hexadecamer (right) obtained at 55 μM further confirmed the cooperative nature of the self-association process (Supplementary Fig. 2j). Consistently, the binding affinity for the tetramer–tetramer

interaction was calculated to be a $K_d$ of 370 μM (68% confidence limits of +130, –80 μM), whereas that of the octamer–octamer interaction showed a $K_d$ of 2.2 μM (68% confidence limits of +1.4, –1.0 μM). These observations suggest that the hexadecamer can readily form once enough octamer is present. Under our experimental conditions, the hexadecamer was the largest reliably detectable complex (Fig. 2b, c), though larger precipitating complexes appeared to form at higher loading concentrations.

To corroborate these findings, interferometric scattering mass spectrometry (iSCAMS)[37] was performed to directly visualize oligomeric distribution of the Cep63 (424–541)•Cep152 (1205–1295) complex in solution. When the complex was freshly diluted to 200 nM and analyzed immediately (i.e., unequilibrated) at pH 5.5, the presence of a broad but presumptive hexadecamer (calculated MW of 207.7 kDa) with an estimated MW of 195 kDa was evident (Fig. 2e, left; green arrow). Further analysis of a third of the acquired images (to reduce the overshadowing tetramer) revealed the presence of a small but distinct 109 kDa complex (a presumed octamer; Fig. 2e, red arrow) hidden under the 58 kDa heterotetramer peak (black arrow) (Fig. 2e, left). When the sample was diluted to 200 nM and left equilibrated at 4 °C, all the higher MW particles disappeared, leaving only the heterotetrameric complex (Fig. 2e, right, and Supplementary Fig. 2k). Time-course analyses revealed that the hexadecameric complex dissociates completely to become a tetrameric building block within 10 min when diluted and left equilibrated (Supplementary Fig. 2l). Taken together, these observations suggest that the tetrameric complex undergoes specific protein–protein interactions to generate concentration-dependent higher MW species and that this process is reversible.

**Identification of a conserved basic Cep152 CC motif critical for preserving the heterotetrameric state of the Cep63•Cep152 building block.** Our numerous efforts to determine the crystal structure of the Cep63 (424–541)•Cep152 (1205–1295) complex were unsuccessful because of the self-assembling activity of the hydrophobic motif (black box in Supplementary Fig. 3a) positioned in the middle of the complex[12]. The crystal structure of the region forming the four-helix bundle (Supplementary Fig. 3a, right) has been reported[12]. Therefore, we next attempted to crystallize various Cep63 fragments upstream of its hydrophobic motif and solved the structure of Cep63 (440–490), which displayed a classical leucine zipper-like dimer whose N-terminal ends project away from each other (Supplementary Fig. 3b, left) (PDB: 7W91; Supplementary Table 4). Comparative analysis of the Cep63 (462–483) region with the previously characterized GCN4 leucine-zipper dimer[38] revealed their structural similarity, each showing pairwise hydrophobic interactions along its

parallelly arranged dimer (Supplementary Fig. 3b, right). Notably, three copies of the Cep63 (440–490) dimer were found in an asymmetric unit (Supplementary Fig. 3a, left) within which two copies appeared to be from crystal-packing artifacts, judging from several lines of evidence. First, the data obtained from SAXS, sedimentation velocity/equilibrium, and iSCAMS analyses (Fig. 2) are all consistent with the dimeric form of Cep63 (440–541). Second, SEC-multi-angle light scattering (MALS) showed that Cep63 (440–490) is mainly (97.9%) in a dimeric state (Supplementary Fig. 3c). Third, the L469A N473R V480K mutations designed to disrupt the parallelly arranged dimeric interactions yielded a delayed peak (12.82 mL) of the monomeric form, whereas the L445A A456K L463K mutations aimed at disrupting the predicted packing interactions did not (Supplementary Fig. 3a, d). Fourth, the Cep63 N-terminus-in and C-terminus-out arrangement observed around a centriole (Fig. 1) does not support a tridirectional protein orientation that a hexameric Cep63 could adopt (Supplementary Fig. 3a, left).

Intriguingly, the Cep63 (440–490) fragment is highly acidic (pI approximately 5.5), raising the possibility that the open-ended N-termini of the Cep63 (440–490) dimer (Supplementary Fig. 3a) could interact with the conserved basic Cep152 (1273–1295) region (pI of 9.5) predicted to form a CC motif (Supplementary Fig. 3e). If this were the case, then the Cep152 (1273–1295) region could substitute the packing interface observed in the Cep63 (440–490) crystal structure. Based on this assumption, we built a helix bundle model by incorporating the crystal structure of the Cep63 (502–541)•Cep152 (1205–1250) complex (PBD: 6CSU) (Supplementary Fig. 3a, bottom) (see Methods for details) and generated mutants either lacking Cep152 (1273–1295) (referred to as "Δ23"; deleting 23 residues) or replacing the two conserved hydrophobic residues, I1279 and L1286 (Supplementary Fig. 3f), with Asp residues (referred to as "2D").

SEC analyses showed that, unlike Cep63 (424–541)•Cep152 (1205–1295) WT, its respective Δ23 (red line in Figs. 3a) and 2D (green line) mutants exhibited substantially faster elution profiles (Fig. 3a and Supplementary 3g), suggesting that mutations in the conserved basic CC region alter the biochemical properties of the complex. Circular dichroism analyses showed that these mutations did not noticeably influence the complex's predominantly α-helical structure (Fig. 3b). Consistently, in sedimentation velocity analyses, the Δ23 mutant appeared as a single 4.14 S species at low concentrations, with an estimated MW of 71 kDa (the best-fit frictional ratio of 1.52) (Fig. 3c), suggesting that it forms a hexamer-like complex (calculated MW of the hexamer is 70.3 kDa). In addition, its capacity to form dosage-dependent higher MW complexes or aggregates (arrows in Fig. 3c) was evident. Analysis with a shorter Δ23 construct [i.e., Cep63 (440–541)•Cep152 (1205–1272)] lacking an additional N-terminal Cep63 (424–439) yielded similar results, showing a sedimentation coefficient of 4.24S, with an estimated molar mass of 64 kDa (calculated MW of the hexamer is 65.2 kDa) (Supplementary Fig. 3h). These findings strongly suggest that loss of the basic CC region (i.e., Δ23) results in formation of a hexamer-like complex at low concentrations. However, analysis of the Cep63 (424–541)•Cep152 (1205–1295) 2D mutant showed the presence of a single species at 4.30S, with a MW of 92 ± 12 kDa (105 kDa based on the interference data) (calculated MW of the octamer is 103.5 kDa) under various concentrations (Fig. 3d). Thus, while the structural details of the 2D mutations are yet to be investigated, the 2D mutant displays a strong propensity to assume an octameric state.

Since the basic CC region appeared to be critical for maintaining the heterotetrameric state of the Cep63 (424–541) •Cep152 (1205–1295) building block, we measured the SAXS data for the Δ23 and 2D mutants at different concentrations. From their respective SAXS data extrapolated to infinite dilution, the DAMMIN envelopes were calculated and compared with the similarly generated envelopes of the heterotetrameric building block shown in Fig. 2a. The SAXS data for the Cep63 (424–541)•Cep152 (1205–1272) (i.e., Δ23) complex showed inter-particle interaction even at lower concentrations (Supplementary Fig. 3i), hence making it difficult to generate a reliable SAXS envelope. To remedy this, data points at the low q region ($q < 0.03 \, \text{Å}^{-1}$) were removed and extrapolated using a linear Guinier fit (Supplementary Fig. 3i, dashed black line). Analysis with the resulting extrapolated data yielded a SAXS envelope morphologically resembling its respective WT, although an increased cross-sectional radius ($Rc = 20.68 \, \text{Å}$ compared to 14.95 Å for WT) was apparent (Fig. 3e, left, and Supplementary Movie 1; Also note somewhat overestimated MWs because of its aggregative nature). The shorter Δ23 construct additionally deleted of the N-terminal Cep63 (424–439) region, which also generated a hexamer-like complex at low concentrations (Supplementary Fig. 3h), yielded a similar cross-sectional radius (Supplementary Fig. 3j, k, and Supplementary Movie 1). Likewise, the Cep63 (424–541)•Cep152 (1205–1295) I1279D L1286D (i.e., 2D) mutant forming an octameric complex (Fig. 3d) yielded an envelope with an increased $Rc$ value (Fig. 3e, right, Supplementary Fig. 3l and Supplementary Movie 1). Guinier analyses of the SAXS data for the Δ23 (extrapolated), shorter Δ23, and 2D mutants extrapolated to zero concentration indicate the absence of aggregation (Supplementary Fig. 3m, left).

SAXS analyses revealed that real-space $P(r)$ interatomic distribution profiles of the Δ23 and 2D mutants (Supplementary Fig. 3m, right) show maximum dimensions ($D_{max}$) slightly larger than that of the WT complex (Fig. 2a). These findings are consistent with the larger sedimentation coefficient values (Fig. 3c, d and Supplementary Fig. 3h) obtained for these mutants that form higher-order oligomers. Overlaid SAXS envelopes showed that these mutants all exhibited a morphology with a bend around the middle of the envelope (Fig. 3e, Supplementary Fig. 3k, and Supplementary Movie 1).

**Atomic force microscopy (AFM) analysis of Cep63 (424–541) •Cep152 (1205–1295) and its octameric 2D mutant**. To corroborate the SAXS-based structural model of the Cep63 (424–541)•Cep152 (1205–1295) and its octameric 2D mutant in Figs. 2a, 3e, we performed AFM analysis under physiological buffer conditions at 5 °C. Interestingly, both complexes displayed a largely elongated morphology but with a less apparent structural kink than that observed from SAXS analyses (Fig. 4a and Supplementary Fig. 4). Since AFM was performed on a functionalized mica surface, the flexibility of the complex could have rendered the complex less kinked than in a solution environment.

Quantification of obtained images revealed that while the lengths of Cep63 (424–541)•Cep152 (1205–1295) WT and its 2D mutant were broadly consistent with those obtained from the SAXS analysis (Figs. 2a and 3e), the heights of the complexes attached to the mica surface were measured to be substantially lower than the cross-sectional diameter measured in solution (i.e., $2 \times Rc$ of the SAXS envelopes provided in Figs. 2a and 3e). Under these conditions, the octameric 2D mutant showed a volume slightly over 2-fold larger than that of the heterotetrameric Cep63 (424–541)•Cep152 (1205–1295) complex (Fig. 4b). Although we were not able to determine how two tetrameric building blocks are arranged to generate an octameric complex at the molecular level, an elongated morphology with a larger volume observed with the 2D mutant suggests that the building block molecules are arranged in a parallel fashion to generate the octameric complex.

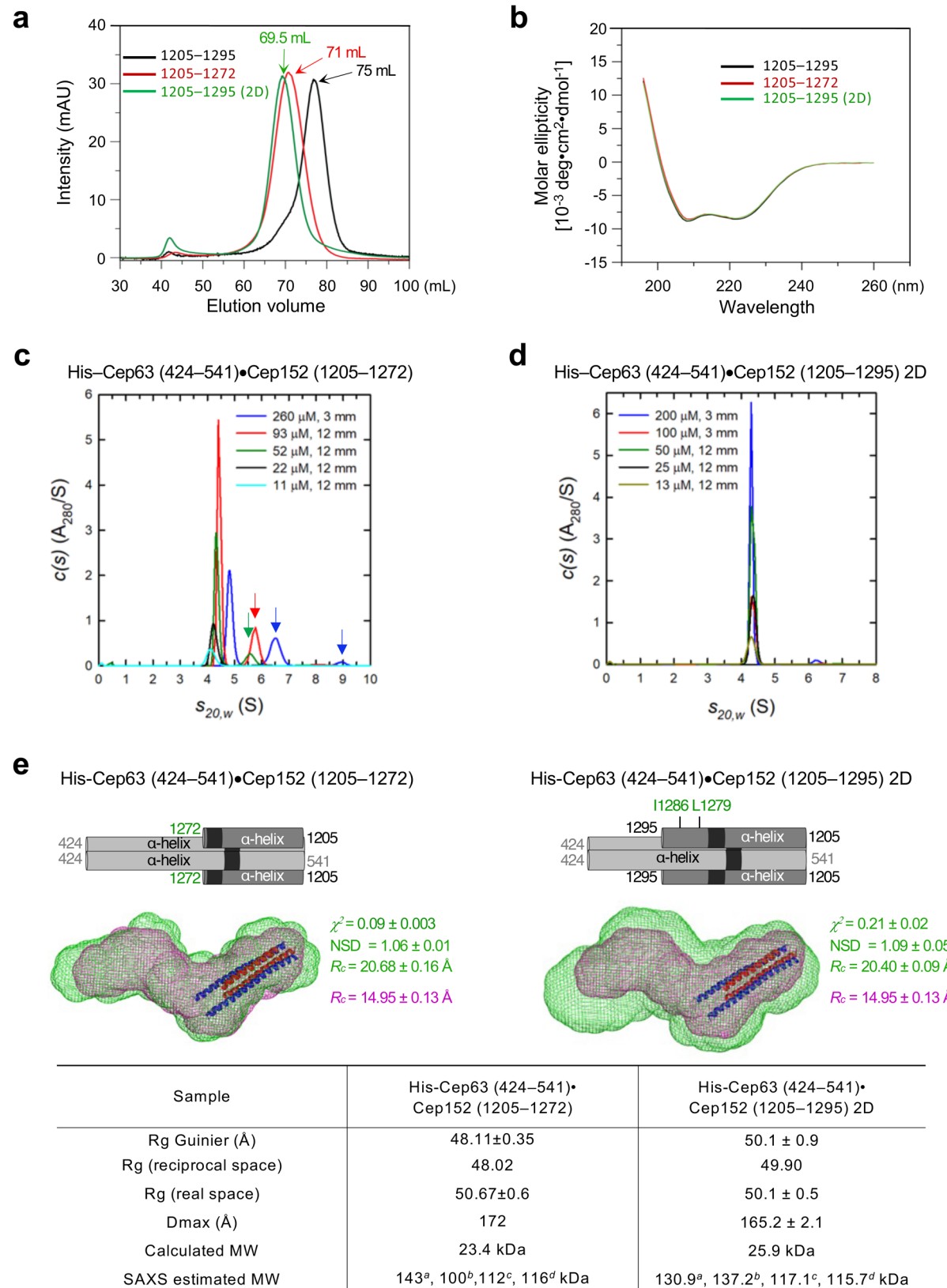

| Sample | His-Cep63 (424–541)• Cep152 (1205–1272) | His-Cep63 (424–541)• Cep152 (1205–1295) 2D |
|---|---|---|
| Rg Guinier (Å) | 48.11±0.35 | 50.1 ± 0.9 |
| Rg (reciprocal space) | 48.02 | 49.90 |
| Rg (real space) | 50.67±0.6 | 50.1 ± 0.5 |
| Dmax (Å) | 172 | 165.2 ± 2.1 |
| Calculated MW | 23.4 kDa | 25.9 kDa |
| SAXS estimated MW | 143[a], 100[b],112[c], 116[d] kDa | 130.9[a], 137.2[b], 117.1[c], 115.7[d] kDa |

**The Cep152 (1273–1295) CC motif is required for Cep63•-Cep152 self-assembly in vitro and Plk4-mediated centriole duplication in vivo.** To determine whether the formation of a heterotetrameric Cep63•Cep152 building block is critical for self-assembling the cylindrical structure, the Cep63 (424–541) •Cep152 (1205–1272) and the Cep63 (424–541)•Cep152

(1205–1295) 2D mutants, which form a hexamer and an octamer, respectively (Fig. 3a–e), were subjected to in vitro self-assembly assays. The assay was performed on a 1–5-kDa poly-L-lysine-coated slide glass by spotting 10 μL of 5 μM purified complex, a concentration comparable to the approximately 16 μM of Cep152 (~270 molecules/centriole[3]) estimated to be in the PCM space

**Fig. 3 Mutations in a conserved basic CC motif of Cep152 prevent the formation of the heterotetrameric Cep63•Cep152 building block. a, b** SEC elution and circular dichroism profiles of the Cep63 (424–541)•Cep152 (1205–1295), Cep63 (424–541)•Cep152 (1205–1272) truncation, and Cep63 (424–541) •Cep152 (1205–1295) 2D complexes. **c, d** Sedimentation velocity $c(s)$ profiles for Cep63 (424–541)•Cep152 (1205–1272) (**c**) and Cep63 (424–541) •Cep152 (1205–1295) 2D (**d**) mutants at various loading concentrations. Samples were analyzed in 3 or 12 mm pathlength cells as indicated. Arrows in (**c**), higher-MW complexes and/or aggregates observed at high concentrations. **e** Schematics and the ab-initio-reconstructed envelopes of the Cep63 (424–541)•Cep152 (1205–1272) truncation (left, green) and the Cep63 (424–541)•Cep152 (1205–1295) 2D mutants (right, green). The envelope of the Cep63 (424–541)•Cep152 (1205–1295) complex (magenta) and the crystal structure (PDB: 6CSU; red and blue helices) of the Cep63 (502–541)•Cep152 (1205–1250) heterotetramer docked in position are overlaid for morphological comparisons. Averaged $\chi^2$ and NSD values calculated from 32 independent DAMMIN reconstructions, along with the $Rc$ calculated from the SAXS data, are shown. Calculated physical parameters from the respective SAXS data of the complexes are given (table). [a,b,c,d], MWs calculated from the SAXS data using four different methods (see Methods for details). Note that due to the aggregative nature of the Cep63 (424–541)•Cep152 (1205–1272) truncation mutant, the SAXS-estimated MWs are somewhat overestimated. 3D-rendered envelopes are provided as Supplementary Movie 1.

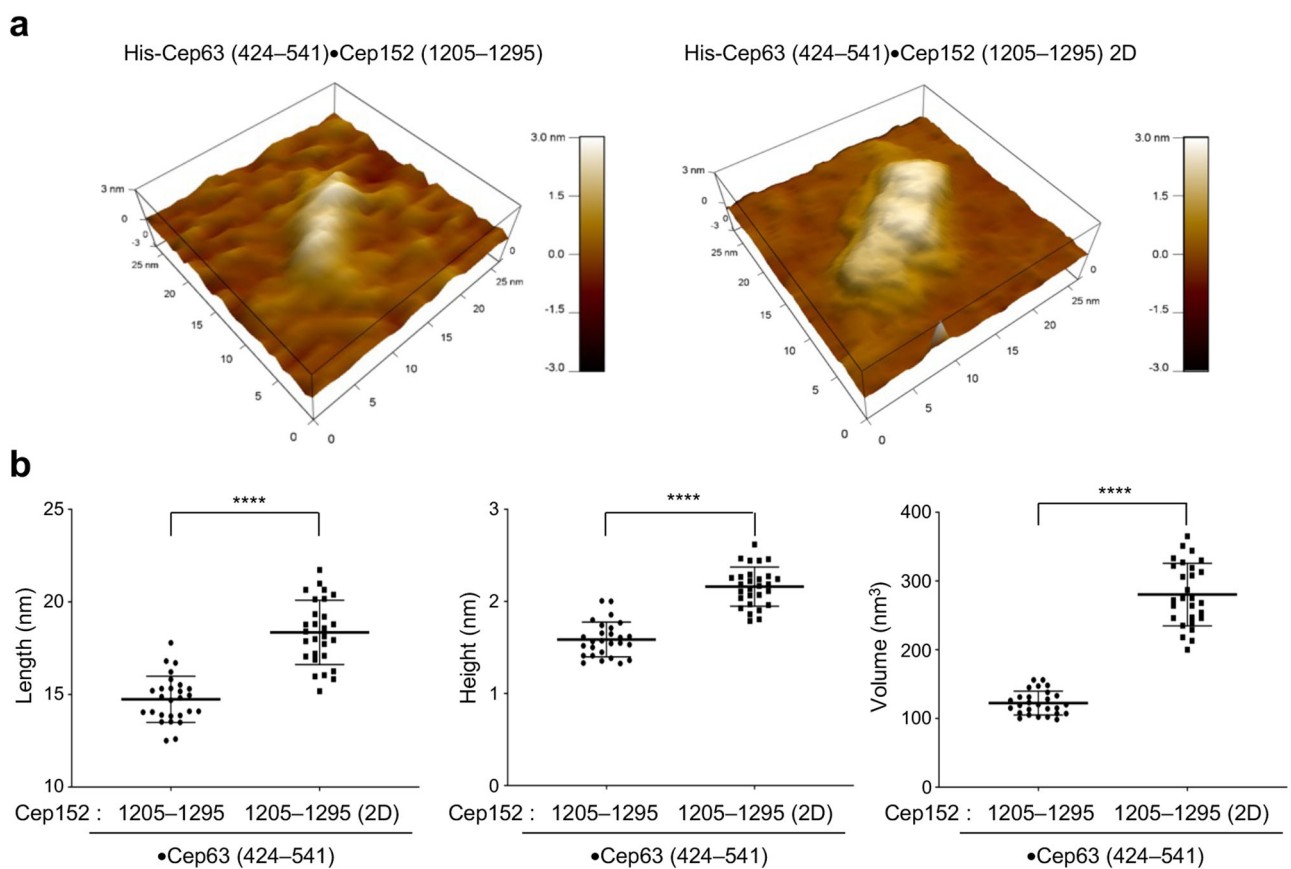

**Fig. 4 Topography of the Cep63•Cep152 complex detected by AFM. a** Surface topographic rendering of the His-Cep63 (424–541)•Cep152 (1205–1295) (left) and His-Cep63 (424–541)•Cep152 (1205–1295) 2D (right) complexes on APS-mica surface. **b** Quantification of the dimension of the two complexes in (**a**) from two independent experiments [for His-Cep63 (424–541)•Cep152 (1205–1295): average length = 14.74 ± 1.25 nm and average height = 1.58 ± 0.18 nm (average volume = 123.32 ± 17.47 nm³); for His-Cep63 (424–541)•Cep152 (1205–1295) 2D: average length = 18.35 ± 1.73 nm and average height = 2.16 ± 0.21 nm (average volume = 280.01 ± 45.38 nm³)]. Bars, mean of n ± s.d. ****$P < 0.0001$ (unpaired two-tailed $t$ test). Note that the lateral dimensions of the particles are tip-broadened, while their height is a precise value. Examples of unprocessed "raw" images are provided in Supplementary Fig. 4.

(Supplementary Fig. 1e). Under these conditions, the corresponding WT complex [i.e., Cep63 (424–541)•Cep152 (1205–1295)] efficiently generated cylindrical self-assemblies, yielding approximately 5078 ± 1171 cylinders per 1 mm² (Fig. 5a, b). While the height of the cylinders reached up to 1 µm, their diameters varied as previously reported[12], indicating structural flexibility. Under these conditions, the Cep63 (424–541)•Cep152 (1205–1272) complex lacking the Cep152 (1273–1295) region (i.e., Δ23) exhibited a severely disrupted ability to generate cylindrical self-assemblies, while the 2D mutant generated cylindrical structures at a greatly reduced level (Fig. 5). Close examination of these assemblies showed they were

morphologically uneven and mostly incomplete, with distorted patches of signals (Supplementary Fig. 5). These findings suggest that the formation of the heterotetrameric building block, which can undergo a reversible tetramer–octamer–hexadecamer transition (Fig. 2 and Supplementary Fig. 2i–l), is a critical step for the Cep63•Cep152 complex to self-assemble into a cylindrical architecture.

We then investigated whether the Cep152 (1273–1295) region plays an important role in the functionality of the protein in vivo using U2OS cells stably expressing RNAi-insensitive, endogenous promoter–controlled Cep152 WT, Cep152Δ(1273–1295) [i.e., Cep152 (Δ23)] or Cep152 I1279D L1286D) [i.e., Cep152 (2D)]

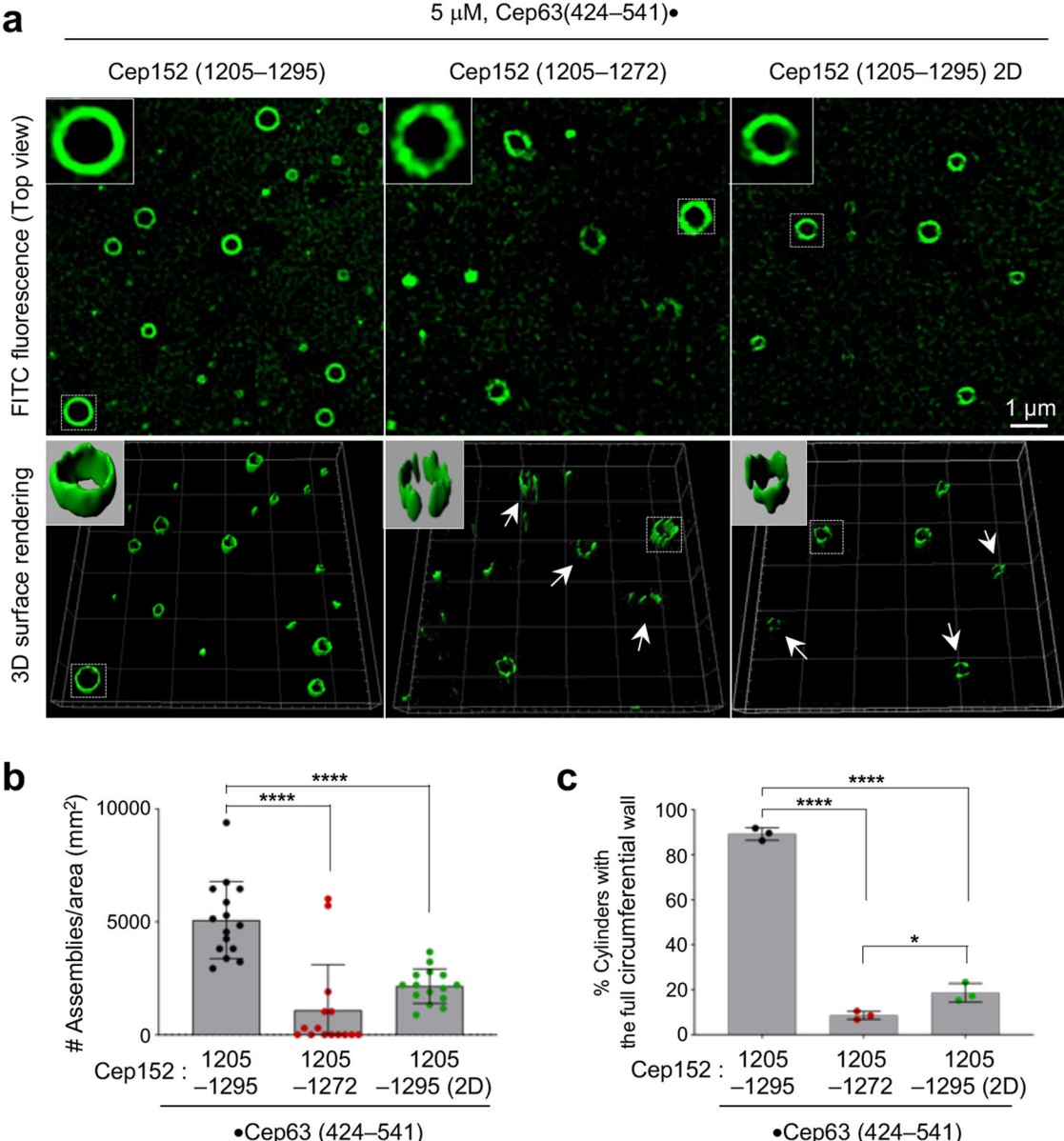

**Fig. 5 3D-SIM (top) and surface rendering (bottom) showing the in vitro self-assemblies generated by the indicated Cep63•Cep152 complex.**
**a** Cylindrical self-assemblies were generated by placing 10 μL of Cep63 (424–541)•Cep152 (1205–1295) (left), Cep63 (424–541)•Cep152 (1205–1272) lacking the 23 basic CC residues (Δ23) (middle), and Cep63 (424–541)•Cep152 (1205–1295) (2D) (right) (5 μM) on a poly-ʟ-lysine-coated coverslip. The assemblies on the coverslip were reacted with FITC, washed, and subjected to 3D-SIM as described in the Methods. Boxes, areas of enlargement; arrows, incomplete cylindrical assemblies. **b** The total number of cylindrical self-assemblies present in the surface area of 1 mm$^2$ was estimated from 15 randomly chosen fields (6813 μm$^2$/field) obtained from three independent experiments ($n = 5$ fields/sample/experiment). ****$P < 0.0001$ (unpaired two-tailed $t$ test). Bars, mean of n ± s.d. **c** The percentage of cylindrical assemblies with a complete circumferential wall quantified from (**b**). *$P < 0.05$, ****$P < 0.0001$ (unpaired two-tailed $t$ test). Bars, mean of three expriments ± s.d.

under Cep152 RNAi conditions (Supplementary Fig. 6a). Immunostaining analyses showed that both Cep152 (Δ23) and Cep152 (2D) mutants displayed apparently normal acetylated tubulin signals along the length of centrioles (Supplementary Fig. 6b, c). Cep57, which is thought to bind to centriolar MT[39,40], also localized around a centriole of these mutants with no detectable defect (Supplementary Fig. 6d). This is somewhat expected since centrosomes in Cep63-deficient mouse spermatocytes display normal centriolar ultrastructure, even though they exhibit abnormal PCM organization[41].

Subsequent analyses revealed that while WT Cep152 efficiently (~83% of the control siGL cells) localized to centrosomes, both the Cep152 (Δ23) and Cep152 (2D) mutants exhibited a

significantly diminished level of centrosome-localized signals (~49% and ~56%, respectively, of the WT) (Fig. 6a, b and Supplementary Fig. 6e). However, 3D-SIM analyses showed that these mutants still displayed a localization dimension (i.e., diameter and height) similar to that of WT (Fig. 6c, d). This could be attributable to the fact that both mutants bear the Cep152(1205–1257) region critical for forming the heterotetrameric bundle with Cep63 (see Fig. 2a), which in turn can interact with Cep57 already localized at the inner region of the PCM[40,42–44]. This notion helps explain why the Cep152 (Δ23) and Cep152 (2D) mutants appear thinned out around a centriole (Fig. 6b) while maintaining their localization dimension in Fig. 6c, d.

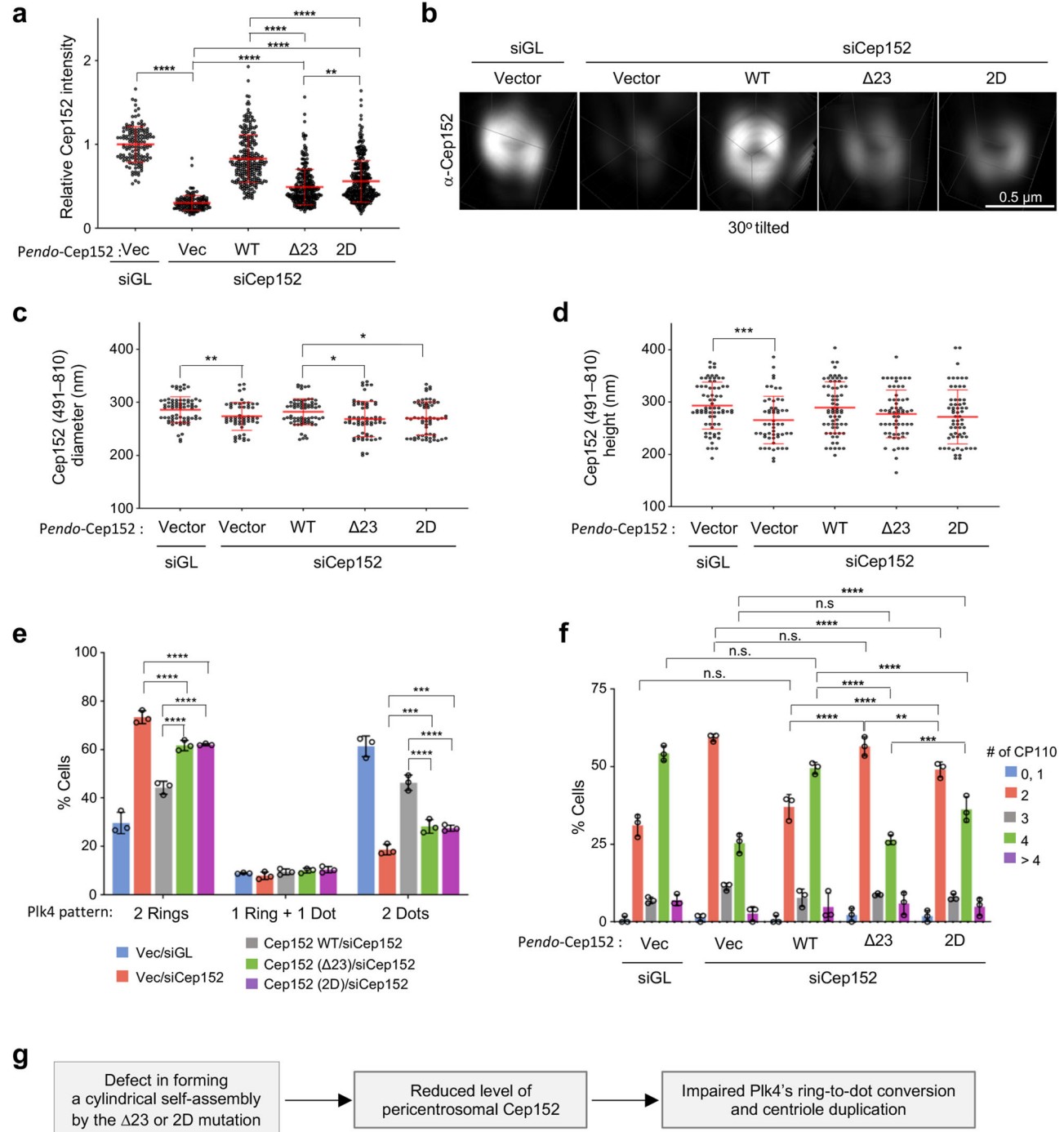

A reduced level of pericentriolar Cep152 could alter the level of Plk4 undergoing concentration-dependent *trans*-autophosphorylation[26] and liquid-liquid phase separation (LLPS), an event critical for Plk4's ring-to-dot relocalization around a centriole and subsequent procentriole formation[28–30]. As expected, Cep152 RNAi cells expressing either Cep152 (Δ23) or Cep152 (2D), which is significantly under-assembled around a centriole (Fig. 6a), exhibited dot-state Plk4 at a considerably lower level than the Cep152 WT-expressing cells (Fig. 6e), even though the amount of centrosome-localized Plk4 in these cells was greater than that of the WT control (Supplementary Fig. 6e, f). Notably, Plk4, which binds to the N-terminus of Cep152[13,14,16], interacted with both Cep152 WT and mutants equally well (Supplementary Fig. 6g). An increased level of centrosome-localized Plk4 in Cep152-depleted cells, which likely occurs through Cep192 binding[16], has been reported[27,45]. These findings suggest that improperly assembled Cep63•Cep152 architecture, which is evident in both Cep63 and Cep152 signals around a centriole (Supplementary Fig. 6h, i, and Fig. 6a, b), fails to properly position Plk4 and promote its ring-to-dot conversion process. Consistent with the decreased Plk4 dot state (Fig. 6e), examination of two downstream centriolar components, Sas6 and CP110, showed that both Cep152 (Δ23) and Cep152 (2D) mutants significantly delayed in duplicating these signals (Fig. 6f and Supplementary Fig. 6h, j, k). Remarkably, enforced elevation of the pericentriolar Cep152 (Δ23) signals to the level of Cep152 WT (this was achieved by transfecting 3-fold higher DNA amount) did not remedy the Cep152 (Δ23)-associated centriole

**Fig. 6 The conserved basic CC motif of Cep152 is required for proper centriole duplication. a–f** U2OS cells stably expressing endogenous promoter (P*endo*)-controlled siCep152-insensitive constructs (shown in Supplementary Fig. 6a) were analyzed after depleting endogenous Cep152. **a** Confocal microscopy analyses for anti-Cep152-immunostained cells obtained from three independent experiments [per experiment, $n \geq 42$ for Vec/siGL (total $n = 140$); $n \geq 50$ for Vec/siCep152 (total $n = 150$); $n \geq 81$ for Cep152/siCep152 (total $n = 256$); $n \geq 86$ for Cep152 ($\Delta$23)/siCep152 (total $n = 273$), $n \geq 108$ for Cep152 (2D)/siCep152 (total $n = 328$)]. **b–f** 3D-SIM analyses for immunostained pericentriolar Cep152, Plk4, and CP110 signals. **b** Representative images showing the localization patterns of Cep152 WT and its respective mutant forms. Cep152's diameters (**c**) and heights (**d**) were determined from the same images obtained from two independent experiments [per experiment, $n \geq 33$ for Vec/siGL (total $n = 70$); $n \geq 27$ for Vec/siCep152 (total $n = 55$); $n \geq 30$ for Cep152/siCep152 (total $n = 68$); $n \geq 31$ for Cep152 ($\Delta$23)/siCep152 (total $n = 63$), $n \geq 30$ for Cep152 (2D)/siCep152 (total $n = 63$)]. Plk4's ring versus dot state (**e**) was quantified from three independent experiments [per experiment, $n \geq 76$ for Vec/siGL (total $n = 258$); $n \geq 66$ for Vec/siCep152 (total $n = 215$); $n \geq 65$ for Cep152/siCep152 (total $n = 234$); $n \geq 67$ for Cep152 ($\Delta$23)/siCep152 (total $n = 244$), $n \geq 71$ for Cep152 (2D)/siCep152 (total $n = 235$)]. CP110 dot numbers (**f**) were quantified from three independent experiments [per experiment, $n \geq 44$ for Vec/siGL (total $n = 144$); $n \geq 50$ for Vec/siCep152 (total $n = 150$); $n \geq 40$ for Cep152/siCep152 (total $n = 129$); $n \geq 43$ for Cep152 ($\Delta$23)/siCep152 (total $n = 136$), $n \geq 54$ for Cep152 (2D)/siCep152 (total $n = 163$)]. Bars for (**a**, **c**, **d**), mean of n ± s.d. Bars for (**e**, **f**), mean of three experiments ± s.d. *$P < 0.05$, **$P < 0.01$, ***$P < 0.001$, ****$P < 0.0001$, n.s., not significant (unpaired two-tailed t test). **g** A summary proposing that a defect in the formation of the Cep63•Cep152 self-assembly (as demonstrated in Fig. 6) results in a reduced level of pericentriolar Cep152 that leads to improper Plk4's ring-to-dot conversion and Plk4-mediated centriole duplication.

---

duplication defect (Supplementary Fig. 6l, m). All the cells exhibiting mislocalized Cep152 ($\Delta$23) signals ($n = 21$) showed only one or two CP110 signals. Flow cytometry analyses revealed that unlike the Cep152 RNAi cells expressing Cep152 (2D), which shows a normal cell-cycle profile, the Cep152 ($\Delta$23)-expressing cells exhibited only a modestly (6.4%) diminished S/G2 population comparable to that of the Cep152 RNAi cells (Supplementary Fig. 6n). Taken together, these data suggest that the inability of the Cep152 ($\Delta$23) or Cep152 (2D) mutant to correctly assemble a cylindrical architecture (Fig. 5) results in reduced Cep152 signals around a centriole (Fig. 6b) that ultimately cripples Plk4-mediated centriole duplication (Fig. 6g).

**Misorganization and improper dynamics of the Cep152 CC mutants defective in forming the heterotetrameric Cep63•-Cep152 building block.** The results shown in Fig. 6 suggest that the Cep152 ($\Delta$23) and Cep152 (2D) mutants incapable of forming the heterotetrameric building block (Fig. 3) fail to properly promote Plk4 function. Since the ability of these mutants to form the cylindrical self-assembly in vitro is crippled (Fig. 5) we further investigated the nature of Cep152 organization around a centriole using MINFLUX nanoscopy, which offers low nanometer-scale precision localization of fluorescent proteins in a 3D space[46,47]. Cells stably expressing siRNA-insensitive, mGFP-Cep152 WT, mGFP-Cep152 ($\Delta$23), or mGFP-Cep152 (2D) were depleted of endogenous Cep152 by RNAi, fixed, and stained with an anti-GFP nanobody fused to a single Alexa Fluor 647 (i.e., one fluorophore per nanobody to avoid overquantifying fluorescence signals) (Fig. 7a–e and Supplementary Fig. 7a–f). Acquired images were filtered and processed to ensure that each localized signal represented an individual protein (Supplementary Fig. 7b, c; see Methods for MINFLUX control analysis with Nup96-SNAP[46]). The resulting data points were then analyzed after discarding outlier signals (Supplementary Fig. 7d, e). Using this approach, we were able to detect hundreds of distinct mGFP-Cep152 molecules per centriole (Fig. 7a, left, and Supplementary Fig. 7e). Although the individual Cep152 signals we detected were likely undercounted due to multiple technical reasons (e.g., antibody detection efficiency, signal quenching, and missing blinks; see Methods for molecule counting details), they were within the range of the data obtained from a targeted proteomics analysis using purified centrosomes (540 copies of Cep152 per centrosome[3]). In addition, the Cep152 molecules were localized around a centriole, displaying a cylinder-like distribution with dimensions similar to those previously reported[4,17] (Fig. 7a, left, and Supplementary Fig. 7c). Therefore, we used these conditions to probe the organizational

features of Cep152 WT and its assembly-defective $\Delta$23 and 2D mutants.

The localization pattern of mGFP-Cep152 WT showed that the signals were randomly distributed within the cylindrical 3D space, with a mean diameter of 384 nm [Note that the N-terminal mGFP diameter is somewhat larger than that of the Cep152 (491–810) epitope (Fig. 6c)] and a mean height of 262 nm (Fig. 7a and Supplementary Fig. 7e; based on the 5th–95th percentile calculation). Most (90%) of the mGFP-Cep152 WT signals were confined within the hollow cylindrical space (i.e., volume), with a radial width of approximately 109 nm (Fig. 7c, d and Supplementary Fig. 7e, f). The wider distribution of mGFP signals could have arisen due to the highly flexible N-terminal region (1–217) of Cep152, which is predicted to be unstructured with undetectable homomerizing activity[12].

Unlike Cep152 WT, both mGFP-Cep152 ($\Delta$23) and Cep152 (2D) mutants displayed clumpy localization patterns skewed toward one or two regions in the PCM (Fig. 7a and Supplementary Fig. 7f). In addition, they spread out, occupying a wider radial space, with an approximately two-fold lower density than that of the WT (Fig. 7c–e and Supplementary Fig. 7e, f). However, like the 3D-SIM–based analysis in Fig. 6d, the heights of their longitudinally localized signals were comparable to those of the WT (Fig. 7a, c and Supplementary Fig. 7e), suggesting that these mutants partially maintain their localization, in part by binding to other PCM proteins, such as Cep57. In line with Fig. 6a, b, the average number of detected mGFP-Cep152 ($\Delta$23) and mGFP-Cep152 (2D) signals (Supplementary Fig. 7e) and the concentrations of these mutants in their respective PCM spaces were significantly reduced (Fig. 7b and Supplementary Fig. 7e). Thus, both Cep152 ($\Delta$23) and Cep152 (2D) mutants fail to properly organize themselves and/or maintain their assemblies within a confined pericentriolar region, consequently disallowing Plk4's concentration-dependent LLPS and induction of centriole duplication (Fig. 6e, f and Supplementary Fig. 6e, f, h, j, k–m).

Consistent with the misorganized and aggregative nature of the Cep152 ($\Delta$23) and Cep152 (2D) mutants, fluorescence recovery after photobleaching (FRAP) analysis revealed that, unlike Cep152 WT, which recovered half of its fluorescence within 30 min post-bleaching, these mutants exhibited a significantly slower recovery rate (Fig. 7f and Supplementary Fig. 7g). This suggests proper formation of the heterotetrameric Cep63•Cep152 building block and progression through the reversible tetramer–octamer–hexadecamer transition are critical steps, not only for self-organizing a pericentriolar architecture, but for maintaining its dynamics to promote Plk4 function.

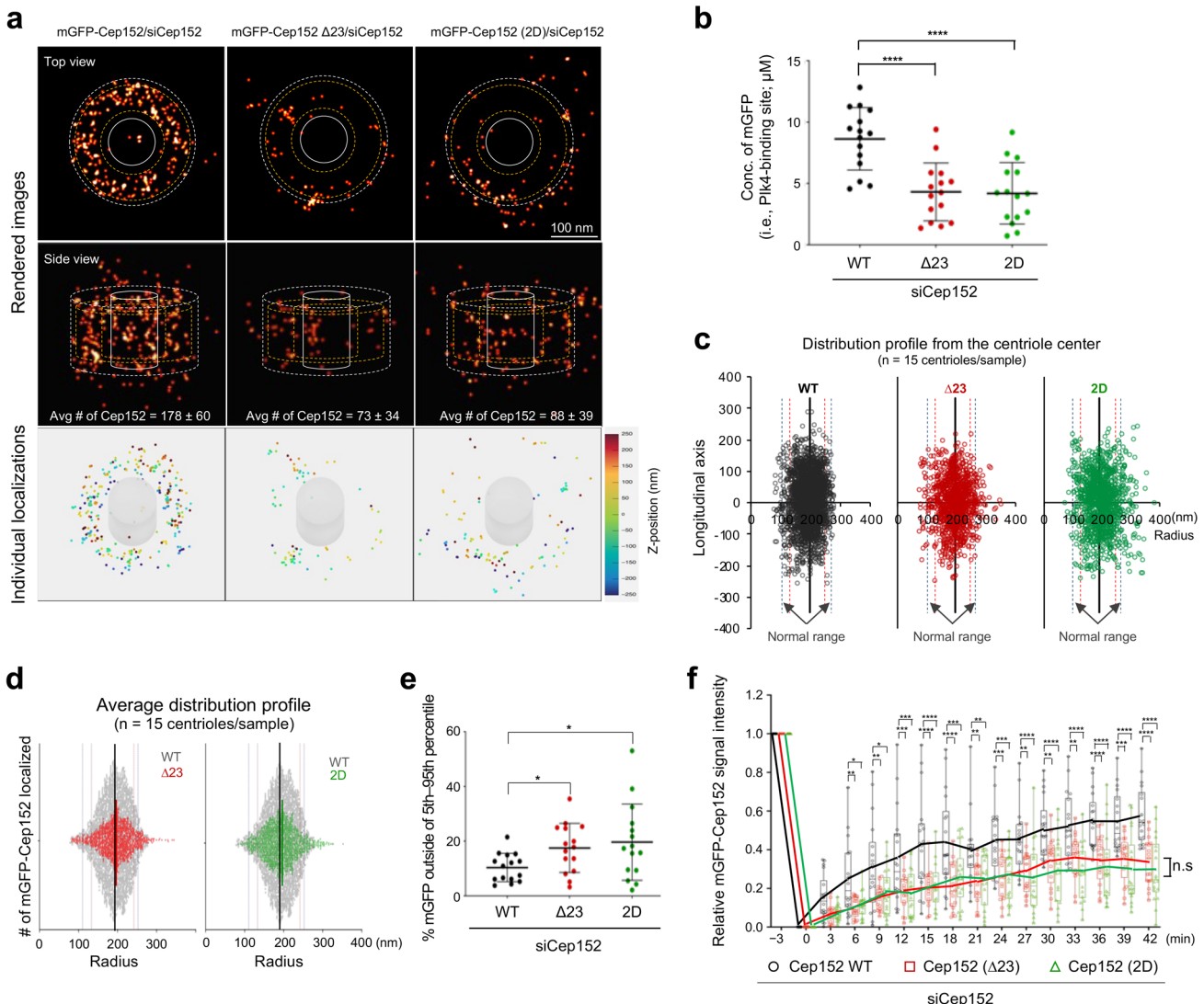

**Fig. 7 Mutations in the conserved CC motif of Cep152 result in misorganizing pericentriolar Cep152. a–e** 3D MINFLUX data for U2OS cells expressing the indicated RNAi-insensitive mGFP-Cep152 constructs (WT, Δ23, and 2D) and depleted of endogenous Cep152 by RNAi. Raw fluorescence signals acquired with an anti-GFP nanobody fused to a single Alexa Fluor 647 (i.e., one fluorophore per nanobody) were filtered and processed as shown in Supplementary Fig. 7b, c (see also Methods). Representative images in (**a**) are shown with dotted boundaries [the maximum (excluding outliers; white dotted line) and the 5th–95th percentiles of the entire minimum-maximum values (yellow dotted lines) of Cep152 WT (see Supplementary Fig. 7d, e)] for easy comparison. Numbers, the radial width and height of the mGFP-Cep152 WT. Concentrations of mGFP-Cep152 in (**b**) were calculated from Supplementary Fig. 7e. Rendered 3D images (top, generated by binning localizations into 0.5 × 0.5 × 0.5 nm voxels, also see Methods), and all localizations plotted in 3D (bottom; respective movies in Supplementary Movie 3) are provided. To reveal mislocalized mGFP-Cep152 signals (**c–e**), the entire min-max range data points shown in Supplementary Fig. 7e were plotted for WT, Δ23, and 2D (**c, d**), and the population outside of the 5th-95th percentiles of the WT radius was determined (**e**). Note in (**d**) that, unlike Cep152 WT (gray dots), both Δ23 an 2D mutants are spread out over a larger area, yielding a greater mislocalized population. All quantifications in (**a–e**), n = 15 centriole images for each group obtained from three independent experiments. Bars in (**b, e**), mean of n ± s.d. *$P < 0.05$, ****$P < 0.0001$ (unpaired two-tailed $t$ test). Vertical lines in (**c, d**), median with 5th-95th (thin red lines) and 1st-99th percentiles (thin blue lines). **f** FRAP analysis for the mGFP-Cep152-mCherry fluorescence localized around a centriole. Images were acquired for 42 minutes at 3-minute intervals. Representative confocal images acquired before and after photobleaching are shown in Supplementary Fig. 7g. Relative signal intensities were quantified from three independent experiments [$n = 18$ for Cep152/siCep152 ($n = 6$/experiment); $n = 17$ for Cep152 (Δ23)/ siCep152 ($n \geq 5$/experiment); $n = 18$ for Cep152 (2D)/siCep152 ($n = 6$/experiment)]. Bars, mean of n ± s.d. *$P < 0.05$, **$P < 0.01$, ***$P < 0.001$, ****$P < 0.0001$ (unpaired two-tailed $t$ test). n.s., not significant.

## Discussion

As a membraneless organelle present in the vast cytosolic space, how the centrosome is organized from hundreds of different proteins while maintaining its dynamic architecture remains unclear. To better understand the assembly and function of an inner PCM structure, we combined biochemical, structural, and cell biological approaches to reveal the molecular basis of Cep63•Cep152 self-assembly formation and investigate the physiological consequences of misorganizing this higher-order platform critical for centriole biogenesis.

**Building a dynamic cylindrical architecture around a centriole via self-assembly processes.** The ability of Cep63•Cep152 to form a cylindrical self-assembly in vitro[12,48], combined with their similar localization patterns in Fig. 1, suggests that these two

proteins self-organize around a centriole by forming a cylinder-like, higher-order structure. Previous work has established that the two contrasting properties of the cylindrical self-assembly—steady-state stability and intrinsic dynamics—are the distinguishable characteristics of a self-organizing system[49,50] [https://doi.org/10.1016/j.tibs.2020.12.011]. Likewise, our data show that, while structurally stable in its self-assembled state, displaying distinct localization patterns (Fig. 7a, c, d), Cep63•Cep152 cylinders undergo a dynamic exchange of their components with those in the surroundings both in vitro[12] and in vivo (Fig. 7f) on a time scale similar to that of keratin and vimentin intermediate filaments[51,52]. In line with this finding, the formation and disassembly of higher MW Cep63•Cep152 complexes were readily reversible in vitro in a concentration-dependent manner (Fig. 2d, e and Supplementary Fig. 2l). These dynamic properties would be important to ensure that molecules in the PCM could freely travel through the cylindrical Cep63•Cep152 structure.

An iSCAMS analysis combined with sedimentation velocity/equilibrium data showed that Cep63 and Cep152 form a highly stable heterotetrameric building block, which then generates higher MW complexes, such as octamers and hexadecamers, as the concentration of the building block complex increases (Figs. 2 and 8a). Notably, while formation of the octameric complex was a low-affinity process ($K_d$ approximately 370 μM), the generation of the hexadecameric complex was a substantially higher affinity process due to the greater number of intermolecular contacts ($K_d$ approximately 2 μM) (Figs. 2d and 8a). Under conditions in which hexadecamer formation was favored (55 μM compared with 12.5 μM), the level of the octameric complex remained low (Fig. 2d), an expected finding when the octamer–hexadecamer transition occurs cooperatively. The low level of dodecamer (which is calculated to be 0.011 μM at 12.6 μM total and 0.26 μM at 55 μM total) is consistent with a simple model in which pairwise tetramer-tetramer contacts compete with the translational entropy cost to build a complex (see Theoretical Models in Methods) (Fig. 8a). We were not able to detect a higher MW complex larger than the hexadecamer under our experimental conditions.

**A conserved basic CC motif in Cep152 safeguards the structural integrity of the heterotetrameric Cep63•Cep152 building block**. Our data showed that the conserved Cep152 (1273–1295) region predicted to form a basic CC motif (Supplementary Fig. 3e) is important for generating the heterotetrameric Cep63 (424–541)•Cep152 (1205–1295) complex (Figs. 2 and 3). Unlike the loss of the Cep152(1273–1295) region (i.e., Δ23), which induces a hexamer-like complex with undiminished self-associating activity (Fig. 3c), mutations of the conserved I1279 and L1286 residues to Asp (i.e., 2D) yielded a stable octameric form with no apparent propensity for self-association (Fig. 3d). Although the defects associated with the Cep152 (Δ23) and Cep152 (2D) mutants are different (i.e., forming a hexameric vs octameric complex, respectively; Fig. 3), the ability to generate cylindrical self-assemblies in vitro was severely defective in both mutants (Fig. 5). These observations suggest that formation of a heterotetrameric complex capable of undergoing reversible tetramer–octamer–hexadecamer transition (Figs. 2, 3 and 8a) is important enough to warrant a thermodynamically favorable assembly process. Not surprisingly, both Cep152 (Δ23) and Cep152 (2D) mutants displayed a grossly misorganized localization pattern around a centriole (Fig. 7a–e and Supplementary Fig. 7d–f) and showed significantly reduced in vivo dynamics, with a poor turnover rate after photobleaching (Fig. 7f). Because of this misorganization, Plk4, which binds to the N-terminus of the Cep152[16], failed to

undergo normal ring-to-dot relocalization and induce procentriole formation (Fig. 6e, f). Collectively, these data suggest that the conserved basic CC motif is required to ensure the structural integrity of the heterotetrameric Cep63•Cep152 building block and the stepwise self-assembly of the cylindrical architecture in a concentration-dependent manner (Fig. 8a).

While our current model (Fig. 8a) suggests that the Cep63 (424–541) and Cep152 (1205–1295) regions play a key role in assembling the cylindrical Cep63•Cep152 platform, other regions of Cep63 and Cep152 could also contribute to the overall organization and functionality of these proteins. It should be noted that the N-terminal Cep63 (1–220) and Cep152 (218–512) regions exhibit significant homomerization activity levels[12], hinting that these regions could add a new layer of organizational complexity to the Cep63•Cep152 architecture around a centriole. Structural analyses of the full-length Cep63•Cep152 complex combined with higher resolution imaging are needed to better comprehend how these two proteins are organized on a larger scale and how they recruit client proteins, such as Plk4, to trigger downstream events.

**A pericentriolar Cep63•Cep152 architecture: A 3D platform for governing Plk4-mediated centriole duplication and beyond**. One prominent feature of the Cep63•Cep152 architecture is that Cep63 and Cep152 form an unusually elongated (approximately 119 nm long, see Fig. 1) building block, which is then radially assembled to generate a cylindrical platform with a wide cross-sectional width around a centriole (Fig. 1). Notably, Cep57, which binds to the N-terminus of Cep63, further expands the platform by localizing to the inner face of the Cep63•Cep152 cylinder[19,40,43]. Cep57 is also reported to interact with the C-terminal PACT domain of PCNT[53]. These findings suggest that the Cep63•Cep152 structure could be a part of a much-extended architectural platform central for organizing the inner PCM. Considering that hundreds of centrosome-localizing proteins have been reported (CentrosomeDB: a human centrosomal proteins database[54]), further investigation into how the Cep63•Cep152 platform is assembled around a centriole could offer new directions in understanding the spatial and organizational regulation of various other proteins in the inner PCM space.

Prior to initiating centriole biogenesis, Cep152-bound Plk4 undertakes its *trans*-autophosphorylation-dependent LLPS and ring-to-dot relocalization at the outskirts of the Cep63•Cep152 platform[28,29,55]. This autonomous process is Plk4 concentration-dependent, as demonstrated previously[31]. Therefore, unlike other soluble kinases undergoing stochastic intermolecular interactions in an open 3D space, the efficiency of Plk4's *trans*-autophosphorylation and LLPS-mediated relocalization to the future procentriole site could be directly impacted by the organizational features of the Cep63•Cep152 architecture (e.g., dimension, physical spacing, and organizational pattern), which would determine the local concentration of Cep152-tethered Plk4. Indeed, MINFLUX imaging of mGFP-Cep152 (Δ23) and mGFP-Cep152 (2D) mutants revealed multiple organizational defects (e.g., reduced Cep152 numbers and spread-out space) (Fig. 7a–e) that severely compromised Plk4's ring-to-dot relocalization and ensuing centriole duplication (Fig. 6). This finding suggests that organizational properties of the Cep63•Cep152 platform function as the crucial determinants for not only recruiting and positioning Plk4 but also timing its activation and induction of centriole biogenesis. A search of human cancer databases revealed multiple somatic mutations clustered within the regions critical for forming the Cep63•Cep152 complex (Catalogue of Somatic Mutations in Cancer, http://www.sanger.ac.uk/cosmic/). Thus, it would be interesting to investigate

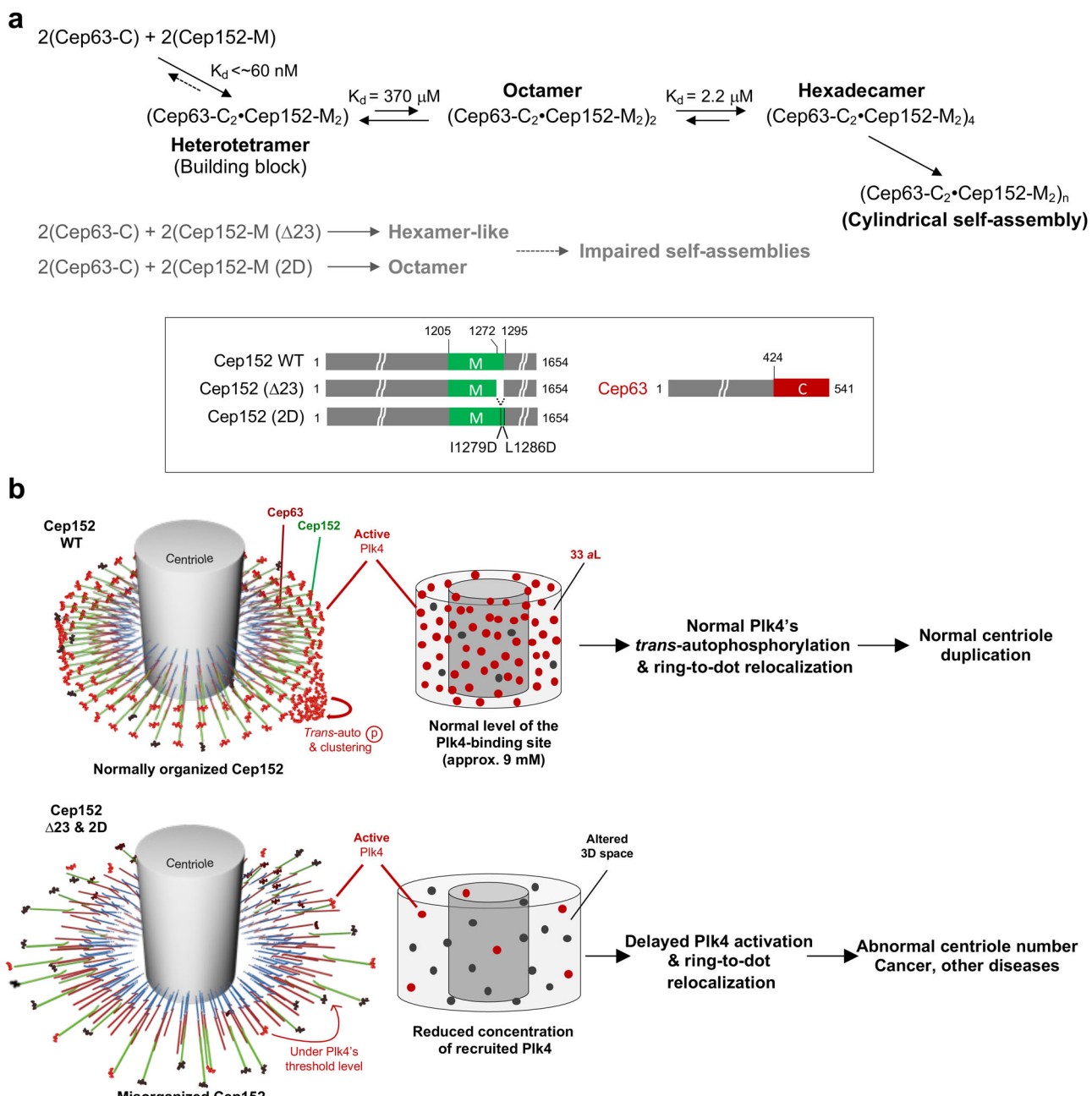

**Fig. 8 Schematics illustrating the stepwise self-assembly processes and the effect of misorganizing the Cep63•Cep152 platform on the Plk4-mediated centriole duplication, the abnormalities of which could lead to the development of various human diseases, including cancer. a** In a concentration-dependent manner, the Cep63 (424–541)•Cep152 (1205–1295) heterotetramer generates octameric and hexadecameric complexes, leading to the formation of a cylindrical self-assembly. iSCAMS was used to determine the $K_d$ value for forming the heterotetrameric building block, while the $K_d$ values for tetramer–octamer and octamer–hexadecamer equilibria were determined from sedimentation equilibrium analyses (Fig. 2d). Formation of the heterotetrameric building block is presumably almost irreversible (dotted arrow), hardly dissociating its components under various conditions. Cylindrical self-assemblies are very stable[12]. The two Cep152 mutants either lacking the (1205–1295) region (Δ23) or containing the I1279D, L1286D mutations (2D) fail to form the heterotetrameric building block and a higher-order self-assembly. **b** Cep152 WT, which localizes within a defined region of the PCM space, properly recruits and promotes Plk4-dependent centriole duplication and cell proliferation. In contrast, its respective Δ23 and 2D mutants misorganize their pericentriolar platform, displaying a broader distribution and lower density of the Plk4-binding Cep152 N-terminus (see text for details). Consequently, these mutants fail to properly promote Plk4's ring-to-dot conversion and centriole duplication. This defect could ultimately lead to various human disorders.

whether these mutations alter the functionality of the Cep63•Cep152 platform and Plk4-mediated centriole duplication.

This work offers new insights into how PCM scaffolds function as a holistic structural entity and how each PCM component coherently operates in the context of a higher-order architecture.

Specifically, a defect in the Cep63•Cep152 self-assembly process results in misorganization of the Cep152 scaffold in the PCM space (Fig. 8), which in turn alters Plk4-dependent centriole biogenesis (Fig. 6). Since improper centriole duplication can lead to chromosome missegregation, aneuploidy, and various human

disorders, including cancer[20,21], proper organization of the Cep63•Cep152 platform would be critical for preventing consequential defects in centrosomal functions. Therefore, given that PCM abnormalities are associated with human disorders, developing an experimental platform to recapitulate the processes of pericentriolar scaffold assemblies would help determine the etiology of centrosome abnormality–induced diseases.

## Methods

**Plasmid constructs**. To generate various lentivirus-based constructs expressing the endogenous promoter (Pendo)-controlled Cep152 WT (pKM7537), Cep152 (Δ23) (pKM7685), and Cep152 (2D) (pKM7684) constructs, a ClaI-SalI fragment containing the respective Pendo-fused Cep152 was cloned into the pHR'.J vector (pKM2994) digested by the same restriction enzymes. The pHR'.J-P$_{CMV}$-based mGFP-Cep152 (pKM7785), mGFP-Cep152 (Δ23) (pKM7786), mGFP-Cep152 (2D) (pKM7787) and mGFP-Cep152-mCherry (pKM 7789) constructs were generated by inserting the respective AscI-PmeI fragment into the pHR'.J-CMV vector (pKM2994) with the same enzymes. The pHR'.J-P$_{CMV}$-mcherry-Cep63-mGFP (pKM 7788) construct was generated by inserting the respective AscI-PmeI fragment into the pHR'.J-CMV vector (pKM2994) digested with AscI-SalI (end-filled). All the Cep63 and Cep152 constructs used for mammalian cell expression contain the respective siCep63- or siCep152-insensitive silent mutations described previously[12].

The pShuttle-CMV vector-based constructs expressing mGFP-Cep152-mCherry (pKM7710), mGFP-Cep152Δ(1273–1295) (i.e., Δ23)-mCherry (pKM7737), and mGFP-Cep152 (I1279D, L1286D) (i.e., 2D)-mCherry (pKM7755) were generated by inserting corresponding BglII-NotI fragments into the pShuttle-CMV vector (Addgene) digested with the same enzymes. The resulting pShuttle-CMV constructs were used to generate pAdEasy-1-based adenoviral constructs (pKM7711, pKM7713, pKM7739, and pKM7756) according to the previously published procedure[56]. The FLAG-Plk4 construct (pKM3445) is reported previously[27].

To generate bacterial constructs dual-expressing Cep63 (424–541) and Cep152 (1205–1295) (pKM5615), Cep63 (220–541) and Cep152 (1140–1295) (pKM6018), Cep63 (219–541) and Cep152 (1205–1295) (pKM5958), Cep63 (424–541) and Cep152 (1205–1272) (pKM5263), Cep63 (440–541) and Cep152 (1205–1272) (pKM5628), or Cep63 (424–541) and Cep152 (1205–1295) 2D (pKM7733), a corresponding Cep63 BamHI-NotI fragment and Cep152 NdeI-XhoI fragment were cloned into the respective sites in pETDuet-1 vector (Novagen). To generate pET28a (Addgene)-(His)$_6$-linked maltose-binding protein (MBP)-tobacco etch virus (TEV) protease–based Cep63 (440−490) (pKM6656), Cep63 (440−490) (L445A, A456K, L463K) (pKM6657), or Cep63 (440−490) (L469A, N473R, V480K) (pKM6658) constructs, the respective Cep63 (440−490) fragments were cloned into the pET28a plasmid (Invitrogen) digested with BamHI and XhoI, then N-terminally tagged with the (His)$_6$-MBP followed by the TEV site.

All the constructs used for this study are summarized in Supplementary Table 1.

**Cell culture and transfection**. Human U2OS (for imaging analyses), HEK293A (for adenovirus production), and HEK293T (for lentivirus production) cells were cultured as recommended by the American Type Culture Collection (ATCC). Transfection was carried out using the calcium phosphate coprecipitation method for lentivirus production, Polyethylenimine Max (PEI Max; Polysciences) for producing AdEasy-1-based adenoviruses, and Lipofectamine RNAiMAX (Thermo Fisher Scientific) for siRNA-based gene silencing.

Depletion of endogenous Cep63 and Cep152 was achieved by transfecting U2OS cells with an siRNA targeting the nt 78–96 region (5'-GGAGCTCATGAA ACAGATT-3') of Cep63[57] and the nt 3099–3119 region (5'-GCGGATCCAA CTGGAAATCTA-3') of Cep152[14], respectively, for 48 h. Where indicated, a luciferase siRNA (siGL; 5'-CGTACGCGGAATACTTCGA-3')[58] was used as a control. All the siRNAs used for this study are listed in Supplementary Table 2.

**Adenovirus and lentivirus production and cell line generation**. Recombinant adenoviruses [mCherry-Cep63-mGFP, mGFP-Cep152-mCherry, mGFP-Cep152 (Δ23)-mCherry, and mGFP-Cep152 (2D)-mCherry] were generated as described previously[56] by transforming the pShuttle constructs (pKM7710, pKM7737, and pKM7755, respectively) into BJ5183 cells (Addgene) for homologous recombination between the pShuttle constructs and the adenoviral pAdEasy-1 vector. Lentiviruses expressing the gene of interest were generated by cotransfecting HEK293T cells with pHR'-CMVΔR8.2Δvpr, pHR'-CMV-VSV-G (protein G of vesicular stomatitis virus), and one of the pHR'.J-CMV-SV-puro-based constructs containing a respective gene. The resulting viruses were used essentially as described previously[59]. U2OS cells infected with the indicated adenoviruses/lentiviruses were then depleted of endogenous Cep63 or Cep152 by RNAi for 48 h before being subjected to further analysis. To maintain the expression of lentivirus-encoded Cep152 WT or mutants, cells were continuously cultured under puromycin (6 µg/mL) selection during the entire period of the experiment.

**Immunostaining**. Immunostainings and subsequent image analyses were performed essentially as described previously[27]. In brief, cells grown on poly-L-lysine (Sigma-Aldrich)-coated No. 1.5 coverslips were fixed with 4% paraformaldehyde, permeabilized with 0.1% Triton X-100 for 5 min at room temperature (RT), and blocked with 5% bovine serum albumin (BSA) in the PBS buffer. The cells were then stained with the indicated primary antibodies (Supplementary Table 3) and appropriate Alexa-fluorophore-conjugated secondary antibodies (Invitrogen). The resulting samples were mounted with ProLong Gold Antifade (Thermo Fisher Scientific) for microscopic imaging. All the antibodies used for this study are listed in Supplementary Table 3.

**Flow cytometry analysis**. For cell cycle analysis, cells stained with 50 µg/mL of propidium iodide (Sigma-Aldrich) were analyzed using BD FACSCalibur flow cytometer and the ModFit LT software (Becton Dickinson).

**Confocal microscopy**. Confocal images were acquired under the confocal mode of the Zeiss ELYRA S1 super-resolution microscope equipped with an Alpha Plan-Apo 63x/1.46 oil objective, 405 nm/488 nm/561 nm/640 nm laser illumination, and standard excitation and emission filter sets. To quantify fluorescence signal intensities and CP110 numbers, images were acquired under the same laser intensities, converted to the maximum-intensity projection of multiple z-stacks, and then analyzed using the Zeiss ZEN v2.1 software (Carl Zeiss Microscopy LLC).

**3D-SIM**. For 3D-SIM analysis, images were acquired using the Zeiss ELYRA S1 microscope above and processed using the ZEN black software (Zeiss). The diameter of cylindrically localized Cep63 and Cep152 was determined by measuring the longest peak-to-peak distance for a given signal. The height of the cylindrical signals was determined by counting the number of z-stacks (110 nm/stack) covering the entire longitudinal length. Due to the intrinsic mechanical limitations of the microscope, all the height estimates have an error range of +/-110 nm. The Zeiss ZEN software enabled us to determine an inter-signal xy-plane distance of up to 33 nm/pixel. Where appropriate, 3D-SIM images were surface-rendered using the Imaris software version 9.2.0 (Bitplane).

**MINFLUX data acquisition and analysis**. For MINFLUX nanoscopy analysis, images were acquired using the MINFLUX 3D system (Abberior, Göttingen, Germany)[46,47]. Briefly, U2OS cells expressing mGFP-Cep152 (see the above lentiviral system) were split onto #1.5 glass coverslip 18-mm round (Warner instruments, 64–0714) and depleted of endogenous Cep152 by RNAi for 2 days. The cells were fixed with 4% paraformaldehyde and decorated with a single domain anti-GFP nanobody fused to a single Alexa Fluor 647 (i.e., one fluorophore per nanobody) (FluoTag®–Q anti–GFP; NanoTag Biotechnologies, Göttingen, Germany).

For MINFLUX imaging, coverslips were treated with a previously described STORM blinking buffer [50 mM Tris–HCl (pH 8.0), 10 mM NaCl, 10% (w/v) glucose, 64 mg/ml catalase (Sigma-Aldrich), 0.4 mg/ml glucose oxidase (Sigma-Aldrich), and 3 mM β-mercaptoethylamine (Sigma-Aldrich)][46]. Fiducials are used to correct for image drifting by pretreating immunostained coverslips with gold nanoparticles (BBI Solution, SKU EM.GC150) for 2–3 min and washing extensively with PBS to eliminate floating gold nanoparticles. The resulting samples were sealed with silicone-glue and subjected to microscopy.

Data from the MINFLUX system consists of coordinates that are grouped together into traces as the system localizes a single fluorescent molecule until it bleaches. We developed criteria to detect individual fluorophores in the image and to separate them from random "noise" localizations of background signals. Any trace that included less than 5 localizations were discarded to remove localizations arising from noise.

Based on the MINFLUX control experiment (below), we assumed that each remaining trace represents a single molecule. For each remaining trace, we determined its position by calculating the weighted mean coordinates (in x, y, and z) of all localizations in the trace with the weight of a localization equal to the ratio of a localization's photon count divided by the total photon count for the trace. The localization precision of these final positions, which we define as the median of the standard deviation in the coordinates for each trace, was ~13 nm.

To obtain the center and mean radius of each centriole shown in Supplementary Fig. 7e, the x–y coordinates were fit to a circle by least-squares. The radial width was calculated in two ways—one from the entire min–max range [defined as the 1.5 × the interquartile range (25th–75th percentiles) in the distribution of distance between the localizations and the central z-axis] and the other from the 5th–95th percentiles of the min–max range (see the Statistical analysis below for detailed information). The height in z was similarly calculated. In this way, we calculated the extent of the cylinder encompassing the centriole. The quantified results are provided in Supplementary Fig. 7e.

3D data detected in MINFLUX are presented as plots of the individual or as pseudo-image renderings based on these localizations. The pseudo-images were generated by binning localizations into 0.5 × 0.5 × 0.5 nm voxels. The resulting image was then convolved with a three-dimensional Gaussian kernel with a standard deviation of 4 nm.

**MINFLUX control experiment with Nup96-SNAP.** To accurately assess the number of mGFP-Cep152 molecules detected under the MINFLUX nanoscopy, multiple potential pitfalls would need to be considered. First, low level of labeling required for MINFLUX may lead to undercounting of molecules. The precise level of under-labeling is hard to estimate for the specimen with unknown molecular number. Second, the molecular number may be estimated incorrectly due to incorrect conversion between localization traces and individual molecules. Individual fluorophores are detected by MINFLUX as clusters with a radius corresponding to the localization precision, containing one or more sets of sequential localizations (traces). A cluster may contain more than one trace because it contains more than one fluorophore, or because the same molecule was detected by more than one trace due to blinking. Third, because of the dimerization capacity and the N-terminal flexibility of mGFP-Cep152, some of the fluorophores detected by anti-GFP nanobody could be closer than the localization precision offered by the MINFLUX. Counting them as one fluorophore will lead to undercounting. On the other hand, counting traces from the same molecule as separate fluorophore will lead to overcounting. Fourth, the blinking behavior, and therefore the probability of localizing a molecule multiple times, is highly dependent on the choice of dye, and again is difficult to assess in a sample that has unknown molecule number and labeling density.

To validate the experimental and data processing methods used in this study, we performed a control MINFLUX experiment using U2OS cells expressing a previously characterized nuclear pore complex (NPC) protein, Nup96, fused to the SNAP tag[46]. The Nup96-SNAP was visualized by Alexa Fluor 647 and data were collected with the same acquisition parameters as those used for the Cep152 samples. In this experiment, localization precision (calculated by taking the standard deviation in position for each trace, and then taking the median of these values) was determined to be approximately 13 nm, a resolution comparable to that reported in the Gwosch et al paper[46]. Removing MINFLUX traces with less than five localizations offered the best compromise between removing traces arising from background and retaining real traces. As reported previously, Nup96-SNAP molecules in the NPC show a two-layered ring morphology with each ring containing eight dimers of Nup96. The individual components of the dimers are separated by only 12 nm and therefore cannot be resolved with MINFLUX in 3D[46]. Therefore, we under-labeled the sample to ensure that for the great majority of dimers only one molecule within the dimer was labeled. By applying the above-described filtering methods, we obtained a total of 81 clusters of localizations (calculated by counting the number of spots observed in the rendered MINFLUX image) from 24 pores observed in confocal. Thus, only 21% of the dimers had at least one label, and we therefore expect that 95.5% $(1-0.21^2)$ of clusters would have a single trace, and we should expect 85 separate localizations from the 81 dimers if each averaged trace is a single fluorophore. In reality, we obtained 93 traces from the 81 dimers. Thus, in our experiments the chance that a single molecule will produce more than one trace is only 9%. Thus, in Cep152 images, we overcount the molecules by no more than 9%. It means that the contribution of under-labeling may be much more prominent, and, most probably, we are underestimating, rather than overestimating the number of Cep152 in a centrosome.

**FRAP analysis.** U2OS cells were cultured on the Lab-Tek II chambered coverglass with a No. 1.5 borosilicate glass bottom (Thermo Fisher Scientific), then treated with 20 nM Cep152 siRNA for 24 h and infected with adenoviruses expressing the indicated mGFP-Cep152 constructs for 14 h. The cells were washed once with phosphate-buffered saline (PBS), replenished with fresh medium, and subjected to time-lapse imaging using a Zeiss LSM880 Airyscan microscope. Centrosome-localized mGFP-Cep152 signals were photobleached with the 75% acousto-optic tunable filter-modulated transmission power of the 488-nm laser at 30 iterations and 3.87 μs pixel dwell time. Fluorescence recovery was monitored by collecting images at three-minute intervals over 42 min. Images were processed using the ZEN 2.3 SP1 software (Carl Zeiss Microscopy LLC), and the processed images were analyzed using the FRAP module of the ZEN software to plot recovery curves.

**Immunoprecipitation and Immunoblotting analyses.** Immunoprecipitation was carried out after lysing transfected cells in TBSN buffer [20 mM Tris-HCl (pH 8.0), 100 mM NaCl, 0.5% Nonidet P-40, 5 mM EGTA, 1.5 mM EDTA], as described previously[27]. Supernatants obtained from centrifuging the lysates at $15,000 \times g$ for 30 min were incubated with anti-GFP antibody (Santa Cruz Biotechnology) for 1 h and further incubated with protein A-agarose beads (Santa Cruz Biotechnology) to precipitate the antibody-associated proteins.

Immunoblotting analysis was carried out by following standard experimental procedures using enhanced chemiluminescence substrate (Thermo Fisher Scientific). Signals were captured using a ChemiDoc™ Imaging System (Bio-Rad Laboratories). All the primary and secondary antibodies used in this work are listed in Supplementary Table 3.

**Protein expression, purification, SEC, and SEC-MALS.** All the proteins except the Cep63 (440–490) constructs (pKM6656, pKM6657, and pKM6658; see below) were expressed in an *E. coli* Rosetta strain (Novagen). To purify the Cep63•Cep152 complexes (pKM5615, pKM5263, pKM5628, and pKM7733), Rosetta cells were cultured in Luria Broth medium at 37 °C until their optical density reached 0.8,

induced with 0.4 mM Isopropyl β-D-1-thiogalactopyranoside overnight at 16 °C, and lysed in an ice-cold buffer [20 mM Tris-HCl (pH 7.5), 150 mM NaCl and 5% (v/v) glycerol, 0.5 mM Tris (2-carboxyethyl) phosphine (TCEP)]. The resulting lysates were then subjected to HisTrap HP column (GE Healthcare) and HiLoad 16/60 Superdex 200 (GE Healthcare) SEC as described previously[12]. The proteins were stored in the final buffer [20 mM Tris-HCl (pH 8.0), 150 mM NaCl, and 0.5 mM TCEP] + 5% (v/v) glycerol at −70 °C until use.

Purification of Cep63 (440–490) WT (pKM6656) and mutants [AKK (pKM6657) and ARK (pKM6658)] was performed using the *E. coli* BL21(DE3) RIL strain (Novagen) cultured at 25 °C for 16 h. Respective lysates were subjected to Ni-NTA affinity chromatography, TEV digestion to cleave off the (His)$_6$-MBP tag, and HiPrep 26/60 Sephacryl S-100 HR (Cytiva) SEC. The protein samples were equilibrated with a buffer containing 20 mM Tris-HCl (pH 8.0), 200 mM NaCl, 5% glycerol, and 2 mM dithiothreitol and stored at −70 °C until use. Selenomethionine-substituted proteins were produced using the *E. coli* B834(DE3) methionine auxotroph strain (Novagen) cultured in a minimal medium.

SEC-MALS was performed using Wyatt's SEC-MALS system to determine the ability of Cep63 (440–490) to form higher-MW complexes. Data collection and analyses were carried out using Astra chromatography software (Wyatt, version 7.1.1.3).

**SAXS data measurement.** SAXS measurements for the respective protein samples were carried out using the 12-ID-B beamline at the Advanced Photon Source at Argonne National Laboratory. Samples were probed with an X-ray beam of 14 keV energy, and X-ray scattering was measured in a 2 M PILATUS detector. The sample-to-detector distance was about 2 meters to achieve scattering $q$ values of $0.005 < q < 0.90$ Å$^{-1}$, where $q$ is the momentum transfer of scattering vector calculated as $q = (4\pi/\lambda)\sin\theta$, and $2\theta$ is the scattering angle, and $\lambda$ is the wavelength of the incident X-ray. For each sample, 30 data frames were recorded in a flow cell with an exposure time of 0.2–1.5 s. The exposure time was selected to obtain good signal-to-noise ratio and avoid radiation damage to the sample. The 2D scattering images were reduced to 1D scattering profiles through the angular integration method using the MATLAB software package at the beamline. The protein scattering profiles were calculated by subtracting the sample scattering profiles from the blank buffer scattering. In order to eliminate inter-particle interactions, concentration series measurements were carried out, then data extrapolated to infinite dilution were used for further data analysis.

**SAXS data analysis.** The radius of gyration Rg and intensity at zero angle I(0) were generated from a Guinier plot in the range of $qR_g < 1.3$. For comparison, Rg and I(0) were also calculated in real and reciprocal spaces using the GNOM program in the $q$ range up to 0.30 Å$^{-1}$[60]. The pair-distance distribution function (PDDF) and maximum dimension ($D_{max}$) were calculated using GNOM. The $D_{max}$ was adjusted until a good fit to the experimental data was obtained and the PDDF curve fell smoothly to zero.

The MWs of the protein complexes were estimated from the scattering data using the Porod volume ($V_{porod}$), apparent volume ($V_{appa}$), correlation volume ($V_c$), and Bayesian statistics methods[61–64].

Individual concentration-independent scattering curves were extrapolated from the SAXS data of each construct. For each scattering curve calculated, 32 ab initio shape reconstructions (molecular envelopes) were generated independently by means of the ATSAS DAMMIN program[33] in slow mode with P2 symmetry (Fig. 2a, Fig. 3a, b). The resulting independent models were aligned, averaged, and filtered by using ATSAS package programs—namely, SUPCOMB, DAMAVER, and DAMFILT—to generate final SAXS envelopes[34]. The reconstructed shapes for every construct were largely similar to one another. Images of SAXS envelopes were generated by the PyMOL Molecular Graphics System, version 3.0 (Schrödinger, LLC).

**Sedimentation velocity analytical ultracentrifugation.** Sedimentation velocity experiments with purified proteins (pKM5615, pKM5263, pKM5628, and pKM7733) were conducted at 50,000 rpm using an An-50 Ti rotor (200,000g at 7.156 cm) on a Beckman Coulter ProteomeLab XL-I analytical ultracentrifuge following standard protocols[65]. Stock solutions of the complexes, along with a matching buffer, were used to prepare a series of solutions at concentrations ranging from 10 to 260 μM, depending on the sample.

Samples of His-Cep63 (424–541)•Cep152 (1205–1295) (pKM5615) were studied at 4 °C in 150 mM NaCl, 20 mM Tris (pH 7.0 or 8.0), 0.5 mM TCEP, and 5% (v/v) glycerol. For studies at pH 5.5, 20 mM citric acid/sodium citrate (pH 5.5) substituted for 20 mM Tris. Samples of His-Cep63 (424–541)•Cep152 (1205–1272) (pKM5263) were studied at 4 °C in 150 mM NaCl, 20 mM Tris (pH 7.5), 10 mM 2-mercaptoethanol, 0.5 mM TCEP, and 5% (v/v) glycerol. Samples of His-Cep63 (440–541)•Cep152(1205–1272) (pKM5628) were studied at 20 °C in 100 mM NaCl, 20 mM Tris (pH 7.5), and 0.5 mM TCEP. Samples of His-Cep63 (424–541) •Cep152 (1205–1295) 2D (pKM7733) were studied at 4 °C in 150 mM NaCl, 20 mM Tris (pH 7.5), 0.5 mM TCEP, and 5% (v/v) glycerol. To minimize possible aggregation, samples studied at 4 °C were kept continuously on ice; loaded into pre-chilled, two-channel, sector-shaped cells that were kept cool; and thermally equilibrated at zero speed in a pre-chilled analytical ultracentrifuge and rotor.

Depending on the loading concentration, 12 mm or 3 mm pathlength centerpieces were used. At 20 °C, zero-speed thermal equilibration was carried out for at least two hours once the rotor reached the temperature in a vacuum. At 4 °C, this equilibration was carried out for at least three hours, with a total time usually exceeding four hours. In all cases, absorbance (280 nm) and interference velocity scans were subsequently acquired at approximately 5-min intervals. Scans were collected overnight, by which time all the material had sedimented, resulting in a baseline contribution. Time-corrected sedimentation data were analyzed in SEDFIT[66] in terms of a continuous c(s) distribution of sedimenting species using a maximum entropy regularization confidence interval of 0.68. The partial specific volumes of the Cep63•Cep152 complexes were determined based on their amino acid composition using SEDNTERP[67]. For samples studied at 4 °C, the solvent density and viscosity were determined experimentally at 20 °C on an Anton Paar DMA 5000 density meter and Anton Paar AMVn rolling ball viscometer, respectively, and corrected to values at 4 °C. Otherwise, the buffer properties were determined using SEDNTERP based on the composition. Sedimentation coefficients were corrected to standard conditions in water at 20 °C ($s_{20,w}$). There was no evidence for sample losses due to aggregation during setup and throughout the experiment.

**Sedimentation equilibrium analytical ultracentrifugation.** Sedimentation equilibrium experiments were carried out to study the self-association of His-Cep63 (424–541)•Cep152 (1205–1295) (pKM5615) at pH 5.5 and 4 °C. Sedimentation equilibrium is the preferred method, as it avoids issues related to the different frictional ratios for the complexes and the slow exchange ($k_{off}$) between oligomers that complicate the interpretation of c(s) profiles. A stock solution of the protein in 150 mM NaCl, 50 mM citric acid/sodium citrate (pH 5.5), 1 mM TCEP, and 5% (v/v) glycerol was used to prepare samples at 12.6 μM, 25 μM, 55 μM, 110 μM, and 220 μM. Samples were loaded in pre-chilled, two-channel, sector-shaped cells and placed into a pre-chilled An50-Ti rotor. Depending on the loading concentration, 12 mm or 3 mm pathlength centerpieces were used. Sedimentation equilibrium experiments were conducted at 4 °C on a Beckman Optima XL-A analytical ultracentrifuge by following standard protocols[65,68]. Data were collected at 3,000, 7000, 11,000, and 20,000 rpm to capture all the sedimenting oligomers by monitoring the absorbance signal at 280 nm and 250 nm. Material losses were noted for the two highest concentrations at 7000 rpm, with the highest concentration indicating the formation of large aggregates even at 3,000 rpm. Samples at 12.6 μM, 25 μM, and 55 μM were well behaved and did not exhibit losses at all the rotor speeds studied, and sedimentation equilibrium at 7000, 11,000, and 20,000 rpm for these concentrations was achieved within 100 h. Equilibrium data collected at different speeds and absorbance wavelengths for the samples at 12.6 μM, 25 μM, and 55 μM were combined and analyzed globally in terms of various self-association models in SEDPHAT[69] with the implementation of mass conservation. Protein partial specific volumes were calculated based on the amino acid composition in SEDNTERP, as were the extinction coefficients at 280 nm. The value of the extinction coefficient at 250 nm was determined as part of the fitting parameters in SEDPHAT. Errors in the self-association equilibrium constants were determined using F-statistics with a confidence level of 68%. In brief, once the best fit was obtained, each equilibrium constant was fixed and scanned stepwise for a series of values above and below the best fit. Fitting was carried out at each step, and the reduced chi-squared values were determined. Values corresponding to 68% confidence levels represent values for the upper and lower limits of the equilibrium constant. The solvent density was determined experimentally at 20 °C on an Anton Paar DMA 5000 density meter and corrected to the value at 4 °C.

Sedimentation velocity and biochemical experiments establish the 2:2 His-Cep63 (424–541)•Cep152 (1205–1295) heterotetramer as the basic unit that self-associates. A qualitative review of the data indicated that the weight-average mass increased with concentration, which was consistent with a reversible self-association. Excellent fits were obtained when data were modeled in terms of a reversible tetramer–octamer (tetramer)$_2$–hexadecamer (tetramer)$_4$ self-association with mass conservation. The tetramer–octamer self-association is characterized by an affinity of 370 μM, whereas a tighter affinity of 2.2 μM characterizes the octamer–hexadecamer equilibrium, which indicates co-operativity. Evidence for co-operativity is noted in the populations of the tetramers, octamers, and hexadecamers. Attempts to fit data using (tetramer)$_1$–(tetramer)$_2$–(tetramer)$_3$, (tetramer)$_1$–(tetramer)$_2$–(tetramer)$_6$, (tetramer)$_1$–(tetramer)$_2$–(tetramer)$_8$, and isodesmic self-association models led to much poorer fits.

**iSCAMS.** The iSCAMS measurement was carried out using the Refeyn TwoMP mass photometer (Refeyn Ltd.). Cleaned coverslips were assembled into flow chambers. All the buffers were filtered through a syringe filter with 0.02 μm pore size (Anotop 10, Whatman). All samples were diluted from stock solutions, and measurements were taken either immediately after diluting the samples (i.e., before reaching an association-dissociation equilibrium) or after keeping the diluted sample at 4 °C for various lengths of time. To detect the hexadecameric Cep63 (424–541)•Cep152 (1205–1295) complex, the sample (110 μM) prepared in a buffer [20 mM sodium citrate, 150 mM NaCl (pH 5.5), and 1 mM TCEP] was diluted to 200 nM and immediately analyzed. To examine the reversible nature of the hexadecameric complex, the sample was diluted to 100 nM, equilibrated for various lengths of time, and analyzed.

**Circular dichroism.** The circular dichroism (CD) spectra for the indicated protein complexes were measured in a Jasco J-715 spectropolarimeter (Jasco International Co.). The proteins were diluted to 0.1 mg/mL in a buffer containing 20 mM sodium phosphate buffer [4.53 mM NaH$_2$PO$_4$, 15.47 mM Na$_2$HPO$_4$ (pH 7.5), 150 mM NaF$_2$, 5% glycerol, and 0.5 mM TCEP]. For each sample, triplicate CD spectra were measured from 195 to 260 nm wavelength using a 0.1-cm pathlength cuvette at the 1 nm spectral bandwidth and 1-second response time. After subtracting the measured buffer spectra, the resulting spectra were averaged, converted into molar ellipticity (deg cm$^2$ dmol$^{-1}$) units, and plotted.

**AFM characterization of complexes.** Freshly cleaved mica substrates were functionalized with a 167 μM solution of 1-(3-aminopropyl)-silatrane (APS; prepared by dilution in pico-pure water [>18 MΩ] from 50 mM stock) for 30 min, rinsed with water, then dried under filtered nitrogen stream as described in Lyubchenko et al. [70]. Protein complexes were diluted to 40 nM in ice-cold buffer [20 mM Tris-Cl (pH 7.8), 150 mM NaCl, 1 mM TCEP] and kept on ice. APS-mica substrates were stored in a petri-dish on ice for at least 15 min before sample deposition. 5 μL of protein solution was deposited on the APS-mica substrate, incubated on ice for 1 min, then rinsed gently with 3 mL of ice-cold buffer. 30 μL of buffer was placed on the sample, and the sample was kept on ice until placed in the AFM.

Topographic images of the protein complexes were collected with a Cypher VRS atomic force microscope (Asylum Research Oxford Instruments, Santa Barbara, CA) outfitted with a heater-cooler stage and a perfusion cantilever holder. Images were collected in AC mode with AC40 (Olympus) cantilevers with free amplitudes of 4.5 nm (0.97 nm/pixel). All imaging was performed in a buffer at 5ºC. Image data were line-by-line flattened to remove tilt from images and rendered in 3D with the Asylum software. The surface of the Cep63 (424–541) •Cep152 (1205–1295) (2D) image (Fig. 4a, right) has been smoothed with a 5 × 5 median filter to remove noise without losing edge information.

The protein AFM images were analyzed using MATLAB (Matlab, MathWorks Inc.) as follows. First, the AFM-measured heights were plotted as a surface map on MATLAB, and individual particles were picked after visual inspection. Second, the image segmentation module from the MATLAB image processing toolbox was used to generate a binary mask image after partitioning each image using a manual thresholding scale. Third, the image region analyzer module was used to calculate the properties of each particle, such as particle length, height, and volume.

**In vitro self-assemblies and their visualization with FITC isomer I.** In vitro self-assemblies were generated essentially as described previously[12] on the surface of a No. 1 coverslip cleaned with a plasma cleaner (Harrick Plasma, Inc.) and argon gas for 1 min, treated with 1 mg/mL poly-L-lysine (1–5 kDa poly-L-lysine, Sigma-Aldrich) for 30 min at RT, washed twice with deionized water, and pre-chilled at 4 °C for 30 min. To form Cep63 (424–541)•Cep152 (1205–1295) complex (5 μM) prepared in the assembly buffer [20 mM Tris-HCl (pH 7.5), 150 mM NaCl, and 0.5 mM TCEP] was placed in spots on the coverslip surface and incubated overnight in a humidified dark chamber. Self-assemblies were visualized by allowing the samples on a coverslip to react with 200 μM FITC isomer I (Sigma-Aldrich) for 1 h at 4 ºC, then decorating the surface Lys residues on the assemblies. The post-reaction samples were then washed twice with the assembly buffer, embedded in SlowFade (Thermo Fisher Scientific), and subjected to 3D-SIM analysis.

**Crystallization, data collection, and structure determination.** Crystals of the Cep63 (440–490) dimer were obtained via the sitting-drop vapor diffusion method at 18 °C after mixing and equilibrating 1 μL of the protein solution (10 mg/mL) with 1 μL of a precipitant solution [600 mM sodium citrate tribasic dihydrate, 100 mM Tris-HCl (pH 7.75), 400 mM 3-(1-methylpiperidinium)-1-propanesulfonate, and 100 mM ethylenediaminetetraacetic acid]. Diffraction data were collected using the beamline 5 °C at the Pohang Accelerator Laboratory, Korea, and processed with the program HKL2000[71]. The complex structure was determined using the single-wavelength anomalous diffraction method in the program AutoSol. The programs Coot[72] and PHENIX[73] were used to build and refine models, respectively. The coordinates and structure factors of the Cep63 (440–490) fragment have been deposited in the Protein Data Bank (PDB: 7W91). The crystallographic data are summarized in Supplementary Table 4.

**Simulation of Cep63 (440–541)•Cep152 (1205–1295).** Based on the hypothesis that the basic Cep152 (1273–1295) CC region could bind to the packing interface of the Cep63 (440–490) crystal, a model with a bent helix bundle (Supplementary Fig. 3a, bottom, 3f) was built by merging two crystal structures [Supplementary Fig. 3a, left (PDB: 7W91, this study) and right (PDB: 6CSU)]. In brief, the Cep152 (1273–1295) region was first aligned with Cep63 (436–457). The Cep152 (1273–1295) structure and its binding to Cep63 were subsequently modeled by mutating the aligned Cep63 (436–457) fragment to Cep152 (1273–1295). The missing linker between Cep152 (1273–1295) and Cep152 (1205–1257) was connected as an alpha helix. Finally, the model was minimized and relaxed by molecular dynamics simulations using the NAMD package[74] and the CHARMM36

force field[75]. The detailed procedure and methods for this structural simulation will be reported elsewhere.

**Statistical analysis and reproducibility**. All the experiments were performed at least three times independently. All values are given as a mean of n ± s.d. *P* values were calculated by unpaired two-tailed *t* test from the mean data of each group. All data reported here are reproducible.

For analyzing MINFLUX data points, box plot analyses were first carried out using Microsoft Excel to determine the minimum and maximum heights and radii (the end of lower and upper whiskers, respectively), which were calculated by $1.5 \times$ the interquartile range (25th–75th percentiles), for each Cep152 WT, Cep152 (Δ23), or Cep152 (2D) group. (The results are provided in Supplementary Fig. 7d.) All the outliers to the minimum and maximum heights and radii (denoted as the percentages in Supplementary Fig. 7d) were excluded from further analysis. All remaining data points within the entire min–max radii were analyzed to yield Supplementary Fig. 7e and the data in Fig. 7b–e. Where indicated, lines corresponding to the mean (middle solid lines), the 1st–99th percentiles (red dotted lines) and the 5th–95th percentiles (blue dotted lines) of the Cep152 WT data points (calculated after excluding the outliers) were provided for easy comparisons among different groups. The height, radius, and volume of the mGFP-Cep152 WT, obtained with the 5th–95th percentiles of the entire min–max range (excluding outliers as defined above) shown in Supplementary Fig. 7e, agree with previous studies[4,12,17]. Therefore, these results were used to determine the volume of the mGFP-Cep152 space and the concentration of the Plk4 binding site (shown in Fig. 7a, b and Supplementary Fig. 7e) located at its N-terminal 1–60 residues[16].

**Theoretical Model of forming higher MW Cep63 (424–541)•Cep152 (1205–1295) complexes**. We have developed a simple model to understand the assembly of tetramers into larger complexes. We model the heterotetramer as a square building block where each surface contains one helix from Cep63 and one from Cep152. Each surface can interact with neighboring tetramers with a free energy ε. As a first step, we consider the formation of octamers from the association of two tetramers. The octamer dissociation constant is

$$k_8 = \frac{c_4^2}{c_8} = c_0 e^{\frac{\epsilon}{k_B T}} \quad (1)$$

where $c_4$ is the free tetramer concentration, $c_8$ is the octamer concentration, and $c_0$ is a reference concentration to fix units. Similarly, we can define the dissociation constant for the hexadecamer

$$k_{16} = \frac{c_4^4}{c_{16}} = c_0^3 e^{\frac{4\epsilon}{k_B T}} \quad (2)$$

Where the exponent reflects the fact that there are four tetramer-tetramer contact energies in the hexadecamer. From the data in Fig. 2 we find that $k_8 = 370$ μM and $k_{16} = 3.0 \times 10^{-13}$ M³. These values can be used to solve for the interaction energy and the reference concentration. The results are ε = −5.13 kT and $c_0 = 63$ mM.

With these two parameters, we can predict the concentration of any size assembly. For instance, the dodecamer concentration can be computed from

$$\frac{c_4^3}{c_{12}} = c_0^2 e^{\frac{2\epsilon}{k_B T}} \quad (3)$$

Which yields values from $c_{12} = 1.1 * 10^{-8}$ M at 12.5 μM to $2.6 * 10^{-7}$ M at 55 μM, which are an order of magnitude less than the hexadecamer concentration at these conditions.

To compute the critical concentration for the macroscopic assembly we balance the chemical potentials of the dilute and the assembled states. In the dilute state the chemical potential is given by the translational entropy $\mu_{dil} = k_B T \ln c_4/c_0$. For the assembled state we note that, neglecting boundary effects, each added tetramer contributes a pair of tetramer-tetramer contacts. Therefore, $\mu_{cyl} = 2\epsilon$ and the saturation concentration of tetramers is given by

$$c_{sat} = c_0 e^{2\epsilon/k_B T} = 2.2 \mu M \quad (4)$$

This value is less than the tetramer concentrations given in Fig. 2. One possibility for this discrepancy is that the solution is supersaturated and a free energy barrier is limiting the formation of larger complexes. To evaluate this possibility, we write the free energy of a square assembly with *L* tetramers on each side.

$$G(L) = 2L(L-1)\epsilon - L^2 k_B T \ln \frac{c_4}{c_0} \quad (5)$$

Where the first term accounts for the attractive tetramer-tetramer contacts and the second term accounts for the loss of translational entropy to bring the tetramers together. Using the parameters found above, this expression predicts that the critical nucleus at 55 μM is a hexadecamer ($L = 2$) with a free energy barrier of about 10 $k_B T$. Therefore, it does not seem plausible that the nucleation barrier accounts for the absence of larger complexes. An alternative possibility is that the pH 5.5 conditions in which the sedimentation experiments were conducted resulted in an excess of positive charge on the tetramers. This would provide a long-range contribution to the interaction energy, not accounted for by the nearest-neighbor ε, that prevented the formation of larger assemblies.

**Reporting summary**. Further information on research design is available in the Nature Portfolio Reporting Summary linked to this article.

## Data availability
The coordinates and structure factors of the Cep63 (440–490) dimer (PBD: 7W91) have been deposited in the Protein Data Bank. The source data for Figs. 1b, 4b, 5b, 5c, 6a, 6c–f, 7b, 7e, and 7f and Supplementary Figs. 6c, 6d, 6f, 6i, 6j, 6l, 6m, 7d, and 7e are provided in Supplementary Data 1. The images shown in Supplementary Fig. 1a, 6a, 6g, and 7a were cropped from the images provided in Supplementary Fig. 8. All other data files are available from the corresponding author upon request.

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

## Acknowledgements
The authors are grateful to Eunhye Lee for preliminary SAXS data, Di Wu and Grzegorz Piszczek for assisting with iSCAMS and CD analyses, Christian Wurm for assisting with MINFLUX microscopy, Catherine Sullenberger and Jadranka Loncarek for sharing unpublished data, Karen Wolcott for assisting flow cytometry analysis, Emilios Dimitriadis for providing APS, and Joe Meyer and Yanling Liu for generating 3D surface-rendered movies. This research was supported by the Intramural Research Program of the National Institutes of Health, National Cancer Institute (K.S.L., Y.-X.W., T.K., R.N.) and NCI Contract No. 75N91019D00024 (W.F.H.), the KRIBB Research Initiative Program (No. KGS1001911), Korea Research Institute of Bioscience and Biotechnology, Republic of Korea (J.I.A.), and an NIH grant R01GM141235 (J.D.S.).

## Author contributions
J.A., L.Z., H.R., K.P.K., Y.Z., L.M., R.G., B.M., D.B., B.K., and K.S.L. designed all the experiments. L.Z. and R.G. carried out pH-dependent sedimentation velocity/equilibrium analyses. L.Z. and Y.Z. performed iSCAMS analyses. J.D.S. performed theoretical analysis and modeling. L.M., H.R., and L.F. conducted SAXS analyses and R.N., S.J.K., and Y.X.W. offered their expertise. H.Y.Y. and B.K. determined the crystal structure (7W91). B.M. performed molecular dynamics simulation and H.R. and K.P.K. analyzed the data. W.F.H. collected atomic force microscopy images and assisted with image analysis. J.A. carried out in vitro assembly assays and all the cell-based analyses. D.B. and T.K. assisted with MINFLUX nanoscopy and developed algorithms for data analysis. J.A. conducted statistical analysis for the MINFLUX images. J.E.P. provided various methodological/

technical assistance. K.S.L. wrote an initial draft and worked with J.A., L.Z., H.R., D.B., T.K., R.G., J.D.S., B.M., B.K., L.F., and W.F.H. to complete the manuscript.

## Funding

## Competing interests
The authors declare no competing interests.
