## [Peer Review File · Communications Biology]

Reviewers' comments:

Reviewer #1 (Remarks to the Author):

The article by Ahn et al. focuses on the structural and functional characterization of the Cep63-Cep152 complex that localizes to the proximal periphery of the centriole and recruits plk4 and thus triggers procentriole formation. Loss of function results in a loss of plk4 localization at the centriole, thus blocking centriole duplication. Understanding how the complex is formed is therefore crucial. This paper follows a previous structural study on this Cep63-cep152 complex published in 2019 (Kim et al. Nat comm). The results presented are consistent with previous findings and thus do not provide much novelty. Nevertheless, the authors here have identified a basic region of the Cep152 protein that appears to be crucial to form the Cep63-Cep152 torus. On the whole, the data seem solid. Nevertheless, there are several points that need to be clarified before publication:

- In the first experiment, the authors mapped the Nter and Cter of Cep63 and Cep152 by expressing tagged versions of these proteins after RNAi depletion (rescue). The result obtained is consistent with the publications of the same group showing that the N-ter in the innermost region and the N-ter of Cep152 is the outermost. What is the level of overexpression relative to the endogenous here? The overexpression may lead to mislocalization with respect to endogenous expression. Can the authors introduce the tag endogenously? Or else antibodies directed against the specific domains?

- The second experiment focuses on the structural interaction of cep63 and Cep152 by studying The Cep63 (424–541)•Cep152 (1205–1295) complex. This complex is very similar to Cep63(502–541)•Cep152(1205–1257) previously studied and resolved by x-ray crystallography (Kim et al. Nat comm 2019). The obtained SAXS map is then used to fit the previously resolved structure of the complex. Here the resolution is very low and it is not clear how the authors oriented the structure inside the map. And is it possible that the complex can make such a large SAXS envelope? The size of the SAXS envelope does not seem to match the sequence of fragments produced to make the complex. Can the authors add the structure prediction also inside? Also it would be interesting to compare with electron microscopy (negative stain). In their previous publications, they had shown that this complex was elongated. How do the authors justify this difference?

- The size exclusion chromatography experiments show that the complex Cep63 (424–541)•Cep152 (1205–1295) is a 2:2 complex. This result is consistent with their previously published work, and not so new. Then the authors observed higher oligomer formation at pH 5.5. What is the justification for doing the experiments at pH 5.5 next? Does this correspond to a biological observation that shows that the torus has a different pH? Can these oligomers be an artifact of in vitro formation? What happens if you assemble your torus in vitro at pH 5.5?

- The authors found an acidic fragment of Cep63 (440–490) that can interact with the basic Cep152 (1273–1295) region. The authors generated point mutations of the two conserved hydrophobic residues, I1279 and L1286, or produce a delta23 lacking the basic region of Cep152 (1273–1295). For the delta23, the authors observed a fastest elution with possibly the formation of a hexamer-like complex. For the 2D mutant, they propose an octameric state. Would it be possible to look at these complexes by negative stain? A model here is also required, it is difficult to understand how these oligomerisation states in a torus context.

- The authors show that the basic Cep152 (1273–1295) CC motif is required for Cep63•Cep152 self-assembly, in vitro, and in vivo. It is clear that the deletion of the basic fragment affects the ability of the protein to form the complex in vitro and in vivo. Did the authors also look at Cep63, in vivo, and in vitro? The authors show that the cep152 torus is less dense and dispersed, is this the case for cep63? The centriole duplication experiment shows that the effect is not very strong. How long did you do the experiment? Is it a rescue for 48h? I could not find the information. And have you checked if the cell cycle is not affected under these conditions?

Reviewer #2 (Remarks to the Author):

Ahn et al. asked how the pericentriolar material (PCM) is structured at a molecular level. In particular they study two coiled-coil PCM proteins, Cep63 and Cep152. The same authors have previously shown that the two proteins self-assemble into a cylindrical structure which may be due to their physicochemical properties to generate high-molecular weight complexes. Cep152 binds to the C-terminal polo-box 1 and 2 of Plk4, the regulator of centriole biogenesis. However, the molecular mechanism of Cep63 and Cep152 self-assembly into a cylindrical structure remains unknown. Their new data show that self-assembly of Cep63 and Cep152 is a stepwise and concentration-dependent process. Based on the use of two binding mutants they show that the basic Cep152 CC-motif (1273-1295) is required for self-assembly and Plk4-dependent centriole duplication.

Major comments:

The manuscript is interesting and very well written. Most of the experiments are carefully performed and are of high quality. However, there are some important experiments missing that are listed below.

Additional experiments:

- The authors show in Fig. 2b that the entire interaction domain between Cep63 and Cep152, aa 424-541(Cep63) and aa1205-1295(Cep152), respectively, seem to bear the full capacity of self-assembly. This needs to be confirmed by full-length protein (either expressed in E.coli or baculovirus) as folding of the very short versions might be affected.
- Did the authors verify that the centriole integrity in siCep152 treated cell (Figs. 1C, D and F) is not affected as Cep152, as Cep152 has a function in centriole duplication? WT Cep152 expression might rescue the defect but maybe not expression of the $\Delta 23$ and DD mutants.
- Do the Cep152 $\Delta 23$ and DD mutants bind less Plk4 in comparison to the WT protein? This needs to be shown using biochemical binding assays.
- To further verify the impact of the basic Cep152 CC motif on centriole duplication it needs to be shown that Sas-6 recruitment to the centrosome is impaired upon expression of Cep152 $\Delta 23$ and DD mutants in cells.
- Does depletion of Cep57 which binds the N-terminal of Cep63 affect centriole duplication when Cep152 is mutated?

Minor comments:

- Fig. 1b: the authors need to comment on why the peak of the mGFP-Cep152 is much higher than in the case of mGFP-Cep63
- Fig. 5a seems a bit blurry and should be improved

Reviewer #3 (Remarks to the Author):

This manuscript by Lee and colleagues examines assembly dynamics of the essential centrosomal protein complex CEP63-CEP152. A key role for this protein complex is the timely recruitment of PLK4 to the centrosome to initiate centriole biogenesis. This work is the continuation of an elegant study by the same group (Nature Comm 2019) where the authors demonstrated that CEP63 and CEP152 self-assemble into a higher-order cylindrical structure that serves as a platform to recruit PLK4. They further showed that interfering with the ability of CEP63 and CEP152 to self-assemble impaired PLK4-dependent centriole duplication.

In this current manuscript the authors expand their previous work by carefully probing the biochemical and biophysical nature of the CEP63-CEP152 interaction with a range of cutting edge multidisciplinary approaches. They find that CEP63-CEP152 form a heterotetrameric complex, which undergoes further stepwise assembly to give rise to the cylindrical structure described

previously. The key advance here is the identification of a 23 amino acid long basic coiled-coil region in CEP152 that is important for heterotetramer formation, and also for centriole duplication.

This is a substantive body of work with carefully designed and well controlled experiments. The quality of data and figures is high throughout the manuscript. While the authors amassed large amounts of data, in particular biophysical and imaging data, whether this study provides new functional insight into the workings of CEP63 and CEP152 is less clear to me. I also have some reservations regarding the conclusions of the paper.

Specific comments:

1. Based on in vitro studies of short fragments of CEP63 and CEP152, the authors conclude that the cylindrical assembly is inherent to the complex rather than a consequence of assembling around the centriolar wall. While the SIM images seem convincing, it is surprising that despite an overall reduction in frequency Delta23 CEP152 mutants can still give rise these structures. In cells, the Delta23 deletion phenocopies CEP152 depletion in terms of centriole duplication defects and therefore the ability of these short fragments to assemble in vitro into cylinders does not fully align with the in vivo functionality of the full-length protein. Indeed, the phenotype observed in cells could have more to do with diminished centrosomal levels of Delta23-CEP152 rather than the lack of the CC domain itself. Pendo-CEP152 intensity was comparable at centrosomes between siCEP152 alone and those expressing Delta23 CEP152 rescue. This is a major caveat of the paper as the novelty of the manuscript lies in the claim that the specific stepwise self-assembly of CEP63 and CEP152 is vital for centriole formation, whereas the in vivo data does not exclude the possibility that the phenotype is purely due to reduced CEP152 levels. I wonder if the authors could devise experiments whereby similar levels of wild-type and mutant CEP152 are recruited to centrosomes and then examine consequences to centriole duplication.
2. The authors mention relevance of CEP63 and CEP152 to pathologies such as cancer, ciliopathy or microcephaly, yet only cite reviews about centrosome biology in general. My literature search suggested existence of disease-linked mutations in CEP63 and CEP152 and thus it would be interesting to mention these in light of the structural insight provided.
3. While Delta23 deletion cannot sustain centriole duplication, from the graph it appears that it only partially disrupts ring to dot conversion of PLK4 when compared to siCEP152 (Fig 5f). The authors could include statistical tests to see if there is indeed some level of rescue.

**Responses to the reviewers' comments:**

In response to the reviewers' comments, we provide the following new data to further strengthen our original manuscript. Point-by-point responses are shown further below.

1. New Fig. 1a, b – As suggested by reviewer #1, we mapped the N- and C-termini of Cep63 and Cep152 by stably expressing mCherry-Cep63-mGFP or mGFP-Cep152-mCherry using a lentiviral expression system. The immunoblot results show that their expression levels are approximately 5–10-fold over their respective endogenous proteins (Note that this level of expression was required to stably detect the mCherry and mGFP fluorescence signals without immunostaining them). The new data are essentially the same as the previous Fig. 1a generated using the adenoviral expression system. These results are described in line 122.
2. New Fig. 4 and Supplementary Fig. 4 – To visualize both the heterotetrameric Cep63 (424–541)•Cep152 (1205–1295) and its respective octameric I1279D L1286D double mutant, we used an atomic force microscopy (AFM)-based analysis under physiological buffer conditions. Measurement of the length and thickness of both tetrameric and octameric complexes corroborated our SAXS data provided in Figs. 2a and 3e. These new data are stated in line 292.
3. New Supplementary Fig. 2c–h – Reviewer #1 questions how we fit the crystal structure of the Cep63(502–541)•Cep152(1205–1257) complex into the SAXS envelop that we developed for the longer Cep63 (424–541)•Cep152 (1205–1295) complex shown in Fig. 2a. To determine the position of the C-terminus of Cep63 within the envelope, we performed SAXS analyses with two additional complexes with different lengths {i.e., His-Cep63 (220–541)•Cep152 (1140–1295) and His-Cep63 (219–541)•Cep152 (1205–1295)}. These newly built envelopes (provided in the new Supplementary Fig. 2e, f), showing a characteristic structural kink due to the presence of two Pro residues (Pro493 and Pro496), enabled us to confidently dock the crystal structure within the envelope shown in Fig. 2a. The overlay of all three envelopes is shown in Supplementary Fig. 2h. The data are described in line 152.
4. New Supplementary Fig. 6h–j – As suggested by reviewers #1 and #2, we carried out additional immunostaining analyses to examine whether Cep63 localization is affected by the Cep152 Δ 23 and Cep152 2D mutations. Indeed, as expected if Cep63 forms a complex with Cep152, Cep63 signals were also significantly diminished in the cells expressing the Cep152 (Δ 23) and Cep152 (2D) mutants. This is stated on line 362. In addition, we found that the Cep152 (Δ 23) or Cep152 (2D)-expressing cells exhibited a significant defect in recruiting Sas6. These results are consistent with Fig. 6f, showing the significantly diminished level of CP110 recruitment in these cells. This new finding is stated along with the CP110 data in line 363.
5. New Supplementary Fig. 6n – As suggested, we performed flow cytometry analyses and concluded that the phenotype associated with the Cep152 (Δ 23) or Cep152 (2D) mutant is not the result of an altered cell cycle. This finding largely aligns with the previous observation that silencing Cep152 by RNAi does not significantly alter the cell cycle (*Kim TS, et al., PNAS 2013*). This is stated in line 369.
6. New Supplementary Fig. 6b–d – To examine the centriole integrity in siCep152 cells, we performed immunostaining analyses to reveal 1) the length and width of centriolar acetylated tubulin signals and 2) the diameter and height of centriole microtubule-binding Cep57 signals. From these analyses, we did not find any significant differences among the control vector, Cep152 WT, Δ 23, or 2D mutant-

expressing Cep152 RNAi cells. These data concur with the previous report that centrosomes lacking Cep63, which works in concert with Cep152 in the inner PCM (Kim TS, et al., Nat Commun, 2019), exhibit normal centriolar ultrastructure, even though they display abnormal PCM organization in Cep63-deficient mouse spermatocytes (Marjanovic M, et al., Nat Commun., 2015). This finding is stated in line 334.

7. New Supplementary Fig. 6g – Coimmunoprecipitation analysis shows that the level of Plk4 bound to Cep152 WT, $\Delta 23$, or 2D mutant is similar to one another. This is expected because Cep152 interacts with Plk4 through its N-terminal region (Cizmecioglu, O. et al., JCB, 2010; Hatch, E. M., et al., JCB, 2010; Kim, TS, et al., PNAS 2013). This finding is stated in line 357.

8. New Supplementary Fig. 6l, m – As suggested by reviewer #3, we compared the level of CP110 recruitment among the siCep152 cells exhibiting comparable levels of pericentriolar Cep152 WT and Cep152 ($\Delta 23$) signals. The results show that Cep152 ($\Delta 23$) still exhibited a similar degree of reduction in CP110 recruitment, when compared with the data in Fig. 6f (the previous Fig. 5f). Furthermore, among the cells exhibiting mislocalized Cep152 ($\Delta 23$) signals, a defect in recruiting CP110 was pronounced. These findings and the in vitro self-assembly data in Fig. 5 suggest that the Cep152 ($\Delta 23$) mutant is defective in forming pericentriolar self-assembly in the first place, and even if it is sufficiently loaded onto the pericentriolar region, its misorganization cripples Plk4-dependent downstream events, such as CP110 recruitment. This point is stated in line 365.

Outlined below are our point-by-point answers to the reviewers' comments.

Reviewer #1 (Remarks to the Author):

The article by Ahn et al. focuses on the structural and functional characterization of the Cep63-Cep152 complex that localizes to the proximal periphery of the centriole and recruits plk4 and thus triggers procentriole formation. Loss of function results in a loss of plk4 localization at the centriole, thus blocking centriole duplication. Understanding how the complex is formed is therefore crucial. This paper follows a previous structural study on this Cep63-cep152 complex published in 2019 (Kim et al. Nat comm). The results presented are consistent with previous findings and thus do not provide much novelty. Nevertheless, the authors here have identified a basic region of the Cep152 protein that appears to be crucial to form the Cep63-Cep152 torus. On the whole, the data seem solid. Nevertheless, there are several points that need to be clarified before publication:

- In the first experiment, the authors mapped the Nter and Cter of Cep63 and Cep152 by expressing tagged versions of these proteins after RNAi depletion (rescue). The result obtained is consistent with the publications of the same group showing that the N-ter in the innermost region and the N-ter of Cep152 is the outermost. What is the level of overexpression relative to the endogenous here? The overexpression may lead to mislocalization with respect to endogenous expression. Can the authors introduce the tag endogenously? Or else antibodies directed against the specific domains?

Response: This reviewer concerns about a potential misbehavior of overexpressed mCherry-Cep63-mGFP and mGFP-Cep152-mCherry by the adenoviral expression system, which tends to express way over endogenous level. Therefore, we used a lentiviral expression system to reduce the expression level. As shown in Supplementary Fig. 1a, b, both proteins are expressed at an acceptable level. Approximately 5–10-fold overexpression was necessary to detect the mCherry and mGFP fluorescence without immunostaining them. Under these conditions, we found that the overall dimension and organization of both Cep63 and Cep152 were similar to those in the original

manuscript. Further lowering the mCherry-Cep63-mGFP and mGFP-Cep152-mCherry expression made it difficult to directly detect the mCherry and mGFP fluorescence.

- The second experiment focuses on the structural interaction of cep63 and Cep152 by studying The Cep63 (424–541)•Cep152 (1205–1295) complex. This complex is very similar to Cep63(502–541)•Cep152(1205–1257) previously studied and resolved by x-rax crystallography (Kim et al. Nat comm 2019). The obtained SAXS map is then used to fit the previously resolved structure of the complex. Here the resolution is very low and it is not clear how the authors oriented the structure inside the map.

Response: To dock the crystal structure of the Cep63(502–541)•Cep152(1205–1257) complex into the SAXS envelop of the Cep63 (424–541)•Cep152 (1205–1295) complex, we used the two Proline residue (Pro493 and P496)-induced structural kink as the morphological landmark. To strengthen our docked model, we performed additional SAXS analyses using two longer complexes [His-Cep63 (220–541)•Cep152 (1140–1295) and His-Cep63 (219–541)•Cep152 (1205–1295)], which allowed us to corroborate the position of the C-terminus of Cep63 (and the N-terminal G1205 of Cep152). The results are provided in the new Supplementary Fig. 2c–g. Using the Pro-induced structural kink, a best-fitting overlay of all three envelopes was generated (as shown in the new Supplementary Fig. 2h). These additional data further support our original docking model shown in Fig. 2a.

And is it possible that the complex can make such a large SAXS envelope? The size of the SAXS envelope does not seem to match the sequence of fragments produced to make the complex. Can the authors add the structure prediction also inside?

Response: It looks like there is some misunderstanding between the size of the SAXS envelope and the length of the crystal structure that we docked. The SAXS envelope was generated with the Cep63 (424–541)•Cep152 (1205–1295) complex, whereas the crystal structure was from a shorter Cep63(502–541)•Cep152(1205–1257) complex, which is approximately half the length. Accurately fitting the empty area with a predicted structure would require an extensive simulation analysis (in my view, this is a good suggestion we could follow in the future).

Also it would be interesting to compare with electron microscopy (negative stain). In their previous publications, they had shown that this complex was elongated. How do the authors justify this difference?

Response: To support our data, we carried out AFM analysis with the tetrameric Cep63 (424–541)•Cep152 (1205–1295) complex and its octameric L1279D L1286D mutant (refer to the data in Fig. 3). The results provided as the new Fig. 4 and Supplementary Fig. 4 show that the dimension of these complexes is consistent with the SAXS data in Figs. 2a and 3e. One noticeable observation is that while the AFM images show a largely straight morphology, the SAXS envelopes show a bent morphology. This could be attributable to the fact that the AFM imaging was performed on a poly-lysine-coated surface, which could potentially alter the morphology (this could be more likely if the protein is flexible, like our complex). In contrast, the SAXS analyses were performed in solution. In addition, the fact that shape reconstructions of elongated particles tend to yield a bent shape (*Volkov VV and Svergun DI, J. Appl. Cryst., 2003*) could have contributed to the morphological differences between the two methods. This is stated in lines 152 and 297 to help reconcile the apparent morphological differences between the AFM result in Fig. 4 and the SAXS envelope in Fig. 2.

- The size exclusion chromatography experiments show that the complex Cep63 (424–541)•Cep152 (1205–1295) is a 2:2 complex. This result is consistent with their previously published work, and not

so new. Then the authors observed higher oligomer formation at pH 5.5. What is the justification for doing the experiments at pH 5.5 next? Does this correspond to a biological observation that shows that the torus has a different pH? Can these oligomers be an artifact of in vitro formation? What happens if you assemble your torus in vitro at pH 5.5?

Response: We found that at pH 5.5, the Cep63 (424–541)•Cep152 (1205–1295) complex becomes less flexible, showing a bell-shaped peak in a Kratky plot analysis for the complex. Thus, in part due to the increased rigidity, the mCherry-Cep63 (424–541)•mGFP-Cep152 (1205–1295) complex formed a cylindrical architecture more efficiently at pH 5.5 than at pH 7.5 [Note that the octamer peak (red arrow in Fig. 2c) is also detectable at pH 7.0, albeit at a reduced level). Thus, since we could detect high M.W. complexes more easily at pH 5.5 than at pH 7.5, we used this condition to easily detect higher MW complexes and investigate the Cep63•Cep152 self-assembly processes.

- The authors found an acidic fragment of Cep63 (440–490) that can interact with the basic Cep152 (1273–1295) region. The authors generated point mutations of the two conserved hydrophobic residues, I1279 and L1286, or produce a delta23 lacking the basic region of Cep152 (1273–1295). For the delta23, the authors observed a fastest elution with possibly the formation of a hexamer-like complex. For the 2D mutant, they propose an octameric state. Would it be possible to look at these complexes by negative stain? A model here is also required, it is difficult to understand how these oligomerisation states in a torus context.

Response: As suggested, we performed AFM analyses with two distinct complexes [Cep63 (424–541)•Cep152 (1205–1295) complex and its octameric L1279D L1286D (2D) mutant] to see if we could directly visualize the morphology of these two complexes. With a high-end AFM instrument that could offer a few-nm resolution, we found that unlike the Cep63 (424–541)•Cep152 (1205–1295) complex, the octameric L1279D L1286D mutant shows a significantly increased overall volume that could accommodate two tetramers (see the new Figure 4 and Supplementary Fig. 4). However, a significantly increased longitudinal length that the 2D mutant displays suggests that the two tetramers are not likely stacked in a completely parallel manner (i.e., the two tetramers could be somewhat staggered). Additional high-resolution structural analyses, such as cryo-electron microscopy, would be required to gain further insights into how the tetramers generate higher MW complexes, such as octamer or hexadecamer, and how they self-assemble into a higher-order cylindrical architecture. In addition, I agree with this reviewer that molecular simulations and mathematical modeling could help better understand the entire self-assembly process. These approaches would undoubtedly require a substantial amount of new work and extensive collaboration, which should be considered beyond the scope of this work.

- The authors show that the basic Cep152 (1273–1295) CC motif is required for Cep63•Cep152 self-assembly, in vitro, and in vivo. It is clear that the deletion of the basic fragment affects the ability of the protein to form the complex in vitro and in vivo. Did the authors also look at Cep63, in vivo, and in vitro? The authors show that the cep152 torus is less dense and dispersed, is this the case for cep63?

Response: As suggested, we conducted an immunostaining analysis using the cells described in Fig. 6a, b. The results provided in Supplementary Fig. 6i show that, indeed, the levels of pericentriolar Cep63 signals in various Cep152 WT and mutants tightly correlate with those for Cep152 signals shown in Fig. 6b. The height of Cep63 signals did not appear to be altered. This is expected because Cep63 binds to the inner layer component, Cep57, which appears normal in Cep152 (Δ 23) and Cep152 (2D) mutants as shown in Supplementary Fig. 6d.

The centriole duplication experiment shows that the effect is not very strong. How long did you do the experiment? Is it a rescue for 48h? I could not find the information. And have you checked if the cell cycle is not affected under these conditions?

Response: The cells stably expressing the indicated constructs were treated with siCep152 for 48 h and subjected to immunostaining analyses. (This information is provided in line 577).

As suggested, we performed flow cytometry analyses. The results show that while Cep152 RNAi cells expressing Cep152 (2D) did not show any apparent cell cycle defect, the Cep152 (Δ 23)-expressing cells exhibited a somewhat (6.4%) lower 4N (S/G2) population, which is comparable to the Cep152 RNAi cells expressing control vector (new Supplementary Fig. 6n). However, the Cep152 (Δ 23) cells still exhibited an essentially unchanged level of CP110 recruitment defect, even after normalizing the data (i.e., adjusting the S/G2 population with the 4 CP110 dot state). See the graph below, for review only! This is stated in line 369.

Reviewer #2 (Remarks to the Author):

Ahn et al. asked how the pericentriolar material (PCM) is structured at a molecular level. In particular they study two coiled-coil PCM proteins, Cep63 and Cep152. The same authors have previously shown that the two proteins self-assemble into a cylindrical structure which may be due to their physiochemical properties to generate high-molecular weight complexes. Cep152 binds to the C-terminal polo-box 1 and 2 of Plk4, the regulator of centriole biogenesis. However, the molecular mechanism of Cep63 and Cep152 self-assembly into a cylindrical structure remains unknown. Their new data show that self-assembly of Cep63 and Cep152 is a stepwise and concentration-dependent process. Based on the use of two binding mutants they show that the basic Cep152 CC-motif (1273-1295) is required for self-assembly and Plk4-dependent centriole duplication.

Major comments:

The manuscript is interesting and very well written. Most of the experiments are carefully performed and are of high quality. However, there are some important experiments missing that are listed below.

Additional experiments:

-The authors show in Fig. 2b that the entire interaction domain between Cep63 and Cep152, aa 424-541(Cep63) and aa1205-1295(Cep152), respectively, seem to bear the full capacity of self-assembly. This needs to be confirmed by full-length protein (either expressed in E.coli or baculovirus) as folding of the very short versions might be affected.

Response: As the editor kindly pointed out in his email letter, expressing and purifying the full-length Cep152 and Cep63 for in vitro studies would be impossible, especially considering the large size of Cep152 (1654 residue-long). However, in our previous publication (Kim TS, et al., Nat Commun, 2019), we used transfected HEK293T cells and clearly demonstrated that Cep63 (424-541) and Cep152 (1205-1295) are necessary and sufficient for the interaction (A copy of the data included below for your information).

-Did the authors verify that the centriole integrity in siCep152 treated cell (Figs. 1C, D and F) is not affected as Cep152, as Cep152 has a function in centriole duplication? WT Cep152 expression might rescue the defect but maybe not expression of the Δ 23 and DD mutants.

Response: It looks like this reviewer meant the data in Fig. 6 (Fig. 5 in the first version). We accessed the integrity of centrioles in the Cep152 (Δ 23) and Cep152 (2D)-expressing cells by quantifying 1) the length and width of acetylated tubulin signals of a centriole and 2) the diameter and height of the

Cep57 protein, which directly interacts with the centriolar microtubules. These results, of which we found no significant differences, are provided in the new Supplementary Fig. 6b–d. These data suggest that while the Cep63-Cep152 complex plays an important role in positioning Plk4 and promoting Plk4-mediated centriole biogenesis, it does not directly contribute to the centriole integrity. These findings are in good agreement with the previous EM data showing Cep63-deficient spermatocytes show normal centriolar ultrastructure (Marjanovic M, et al., Nat Commun, 2015). The data are described in line 334.

-Do the Cep152 Δ23 and DD mutants bind less Plk4 in comparison to the WT protein? This needs to be shown using biochemical binding assays.

Response: As suggested, we performed a coimmunoprecipitation analysis using transfected HEK293T cells. The results show that the Cep152 (Δ23) and Cep152 (2D) mutants bind to Plk4 as effectively as their respective WT. This result is expected since the N-terminal fragment (residues 1–60) of Cep152 interacts with Plk4 cryptic polo-box (CPB) (Cizmecioglu, O. et al., JCB, 2010; Hatch, E. M., et al., JCB, 2010; Kim, TS, et al., PNAS 2013). The crystal structure of the Cep152 (1–60)•Plk4 CPB complex is reported (Park, SY, et al., Nat Str Mol Biol, 2014).

-To further verify the impact of the basic Cep152 CC motif on centriole duplication it needs to be shown that Sas-6 recruitment to the centrosome is impaired upon expression of Cep152 Δ23 and DD mutants in cells.

Response: As suggested, we performed immunostaining analysis using the cells described in Fig. 6 and Supplementary Fig. 6a. Results showed that the Cep152 (Δ23) and Cep152 (2D) mutants, defective in recruiting CP110 (Fig. 6f and Supplementary Fig. 6k), are also similarly defective in recruiting Sas6. These data are provided in the new Supplementary Fig. 6h, j. This finding is stated in line 363.

-Does depletion of Cep57 which binds the N-terminal of Cep63 affect centriole duplication when Cep152 is mutated?

Response: It is well documented in the field that Cep57 functions upstream of Cep63 and Cep152 (Lukinavicius G, et al., Curr Biol, 2013; Wei Z, et al., MCB, 2020; Zhao H, et al., JCS 2020; Ito KK, et al., JCB, 2021). What is somewhat controversial is the role of Cep57. While our data show that Cep57 plays a major role in recruiting two downstream components, Cep63 and Cep152 (as evidenced by severely delocalized Cep63 and Cep152 signals in Cep57-/- MEF cells) (Wei Z, et al., MCB 2020), two other papers show almost equal functional redundancy of these two proteins. In all cases, however, the depletion of Cep57 is expected to delocalize Cep152, its downstream component. The suggested experiment could be carried out as a part of a detailed and comprehensive study aimed at clarifying the epistasis of the Cep57/Cep57L1-Cep63-Cep152 axis. In this study, we are focusing on investigating the effect of misorganized Cep152 on the Plk4-dependent centriole biogenesis.

Minor comments:

- Fig. 1b: the authors need to comment on why the peak of the mGFP-Cep152 is much higher than in the case of mGFP-Cep63

Response: There might be some misunderstanding. The graph in Fig. 1b shows that the “diameter” of the mGFP fluorescence from mGFP-Cep152-mCherry is 350 nm, whereas that of the mGFP fluorescence from mCherry-Cep63-mGFP is 278 nm. The quantification was carried out to show that both Cep63 and Cep162 are anti-parallelly organized in a way that their total length of the complex

(i.e., from the N-terminus of Cep63 to the N-terminus of Cep152) could horizontally (i.e., 90° from the longitudinal axis of a centriole) extend up to 119 nm (as depicted in Fig. 1c).

- Fig. 5a seems a bit blurry and should be improved

Response: This is now fixed using black circles rather than gray circles.

Reviewer #3 (Remarks to the Author):

This manuscript by Lee and colleagues examines assembly dynamics of the essential centrosomal protein complex CEP63-CEP152. A key role for this protein complex is the timely recruitment of PLK4 to the centrosome to initiate centriole biogenesis. This work is the continuation of an elegant study by the same group (Nature Comm 2019) where the authors demonstrated that CEP63 and CEP152 self-assemble into a higher-order cylindrical structure that serves as a platform to recruit PLK4. They further showed that interfering with the ability of CEP63 and CEP152 to self-assemble impaired PLK4-dependent centriole duplication.

In this current manuscript the authors expand their previous work by carefully probing the biochemical and biophysical nature of the CEP63-CEP152 interaction with a range of cutting edge multidisciplinary approaches. They find that CEP63-CEP152 form a heterotetrameric complex, which undergoes further stepwise assembly to give rise to the cylindrical structure described previously. The key advance here is the identification of a 23 amino acid long basic coiled-coil region in CEP152 that is important for heterotetramer formation, and also for centriole duplication.

This is a substantive body of work with carefully designed and well controlled experiments. The quality of data and figures is high throughout the manuscript. While the authors amassed large amounts of data, in particular biophysical and imaging data, whether this study provides new functional insight into the workings of CEP63 and CEP152 is less clear to me. I also have some reservations regarding the conclusions of the paper.

Specific comments:

1. Based on in vitro studies of short fragments of CEP63 and CEP152, the authors conclude that the cylindrical assembly is inherent to the complex rather than a consequence of assembling around the centriolar wall. While the SIM images seem convincing, it is surprising that despite an overall reduction in frequency Delta23 CEP152 mutants can still give rise these structures. In cells, the Delta23 deletion phenocopies CEP152 depletion in terms of centriole duplication defects and therefore the ability of these short fragments to assemble in vitro into cylinders does not fully align with the in vivo functionality of the full-length protein. Indeed, the phenotype observed in cells could have more to do with diminished centrosomal levels of Delta23-CEP152 rather than the lack of the CC domain itself. Pendo-CEP152 intensity was comparable at centrosomes between siCEP152 alone and those expressing Delta23 CEP152 rescue. This is a major caveat of the paper as the novelty of the manuscript lies in the claim that the specific stepwise self-assembly of CEP63 and CEP152 is vital for centriole formation, whereas the in vivo data does not exclude the possibility that the phenotype is purely due to reduced CEP152 levels. I wonder if the authors could devise experiments whereby similar levels of wild-type and mutant CEP152 are recruited to centrosomes and then examine consequences to centriole duplication.

Response: As the reviewer noted, the Cep152 ($\Delta 23$) mutant can still significantly localize around a centriole (Fig. 6a), even though the complex lacking the Cep152 (1273–1295) region [i.e., the Cep152 (1205–1272) •Cep63 (424–541) complex] is severely defective in forming cylindrical self-assemblies (Fig. 5; former Fig. 4). This could be attributable to the fact that the Cep152 ($\Delta 23$) mutant can still effectively interact with Cep63, which can in turn target the Cep63•Cep152 ($\Delta 23$) complex to the centriolar MT-bound Cep57. [Note that the N-terminus of Cep63 interacts with the N-terminal region of Cep57 by forming a four-helical bundle (Wei Z, et al., MCB, 2020; Zhao H, et al., JCS 2020)]. This may help explain how the self-assembly-defective Cep152 ($\Delta 23$) mutant can be partially recruited to the Cep57-loaded centriole with a diameter and height similar to those of Cep152 WT. This possibility is described in line 345.

This reviewer asks an important question as to whether increasing the centrosome-associated Cep152($\Delta 23$) level could rescue its defect in Plk4-dependent centriole duplication. If proper assembly and organization of Cep152 were critical for promoting Plk4-mediated downstream events, then the Cep152($\Delta 23$) mutant, which is defective in forming pericentriolar self-assembly, could exhibit centriole duplication defect even after remedying its reduced pericentriolar localization. To address this question, we performed immunostaining analyses with U2OS cells transfected with endogenous promoter-controlled (*Pendo*) Cep152 WT or Cep152 ($\Delta 23$). To reach a similar level of centrosome-associated Cep152 ($\Delta 23$), 3-fold more DNA was used for transfection. Under these conditions, Cep152 signal intensities and CP110 dot numbers were simultaneously quantified. The results provided in Supplementary Fig. 6l, m show that while approximately 54% of the cells expressing Cep152 WT exhibited 4 CP110 dot signals, still only 28% of the Cep152 ($\Delta 23$) cells displayed the 4 CP110 dot state. In addition, we did not observe any correlation between the level of Cep152 ($\Delta 23$) signals and the number of CP110 dots. These new data consolidate our notion that a defect in forming cylindrical self-assembly by the $\Delta 23$ (or 2D) mutation results in a reduced level of pericentriolar CP110. This defect ultimately leads to delayed Plk4 ring-to-dot conversion and centriole duplication. This view is summarized in Fig. 6g. These findings are described in line 365.

2. The authors mention relevance of CEP63 and CEP152 to pathologies such as cancer, ciliopathy or microcephaly, yet only cite reviews about centrosome biology in general. My literature search suggested existence of disease-linked mutations in CEP63 and CEP152 and thus it would be interesting to mention these in light of the structural insight provided.

Response: The microcephaly mutations reported (Kalay E, Nat Genetics, 2011; Sir .J. et al, Nat Genetics, 2011; Guernsey DL, et al., Am. J. Hum. Genet, 2010) are all present outside of the Cep63 (424–541)•Cep152 (1205–1295) complex studied here. However, a search of human mutation databases identified from cancer patient tissues (Catalogue of Somatic Mutations in Cancer, <http://www.sanger.ac.uk/cosmic/>) revealed somewhat clustered somatic mutations in the regions critical for forming the Cep63•Cep152 complex. Systematic analyses of these mutations (although the numbers are too many with no apparent hot spot mutations) could be worthwhile to investigate. This is mentioned in line 519.

3. While Delta23 deletion cannot sustain centriole duplication, from the graph it appears that it only partially disrupts ring to dot conversion of PLK4 when compared to siCEP152 (Fig 5f). The authors could include statistical tests to see if there is indeed some level of rescue.

Response: As suggested, we carried out statistical analysis to see if the Cep152 ($\Delta 23$) or Cep152 (2D) mutant rescues the CP110 recruitment defect to some degree. In this analysis, we found that, indeed, the Cep152 (2D) mutant, but not the more disruptive Cep152 ($\Delta 23$) mutant, showed a

statistically significant rescue of the CP110 recruitment. Similar data were obtained when Sas6 was analyzed (see the new Supplementary Fig. 6j). These observations are in line with the data in Fig. 5 that show that the ($\Delta 23$) mutation is more disruptive than the 2D mutation in self-assembling the cylindrical architecture in vitro.

Sincerely,

Kyung Lee

Figure 1a from *Kim TS, et al., Nat Commun (2019)*.

Data adjusted from Fig. 6f (Review only)

REVIEWERS' COMMENTS:

Reviewer #1 (Remarks to the Author):

The authors have answered the points I raised in my initial assessment. I'm very pleased with the work they've done to address them. I would like to thank the authors for their work. I, therefore, support the publication of this article.

Reviewer #2 (Remarks to the Author):

In the revised version of the manuscript, the authors have addressed most main issues raised in the previous review.

However, centrosome integrity might be still affected in response to the time used for siRNA treatment of cells.

I am in favor of publication of the article in Communications Biology.

Reviewer #3 (Remarks to the Author):

The authors have generated new data and analysis, and these addressed my comments in a satisfactory manner.